# Identification of antimicrobial peptides from ancient gut microbiomes

Sizhe Chen [1,2,15], Yue Yuan[3,15], Yun Wang[1,2], Ye Peng [1,4,5], Hein Min Tun [1,4,5], Zhimin Jiang [6], Yinglei Miao [3], Sunjae Lee [7], Xiaole Yin [8], Xiaotao Shen[9], Orlando DeLeon[10], Eugene B. Chang[10], Francis Ka Leung Chan[1,11,12,13], Yang Sun [3] ✉, Siew Chien Ng [1,2,14] ✉ & Qi Su [1,2] ✉

Fecal coprolites preserve ancient microbiomes and are a potential source of extinct but highly efficacious antimicrobial peptides (AMPs). Here, we develop AMPLiT (AMP Lightweight Identification Tool), an efficient tool deployable to portable hardware for AMP screening in metagenomic datasets. AMPLiT demonstrates AUPRC performances of $0.9486 \pm 0.0003$ and reasonable overall training time of $3200 \pm 53$ s. By computationally utilizing AMPLiT, we analyze seven ancient human coprolite metagenomes, identifying 160 AMP candidates. Of 40 representative peptides synthesized, 36 (90%) peptides demonstrate measurable antimicrobial activity at 100 μM or less in vitro. Strikingly, approximately two-thirds of these peptides are sourced from *Segatella copri*, a dominant ancient gut commensal that is conspicuously underrepresented in modern populations, particularly those with Westernized lifestyles. Representative *S. copri*-derived AMPs exhibit disruptions against membranes of pathogenic bacteria, coupled with low cytotoxicity and hemolytic risk. In vivo, lead peptides demonstrate potent antibacterial and wound-healing efficacy comparable to traditional antibiotics, especially in combating gram-positive pathogens. Our findings highlight the ancient gut microbiomes as sources of novel AMPs, offering valuable insights into the historical role of *S. copri* in human health and its decline in contemporary populations.

The gut microbiome is increasingly recognized as a vast repository of biologically active antimicrobial peptides (AMPs), offering a critical resource for addressing the global antibiotic resistance crisis[1–4]. Beyond their role in modulating and maintaining host microbial community equilibrium[5], AMPs employ diverse mechanisms—such as membrane disruption, intracellular division inhibition, and immunomodulation to suppress pathogens, thereby enhancing the adaptive resilience of hosts[6,7]. However, contemporary research remains disproportionately focused on AMPs from modern human microbiomes[1,3], which coexist with pathogens that have evolved sophisticated resistance mechanisms[8]. This arms race underscores the

limitations of mining modern microbiomes for novel AMPs, as pathogens may be equipped with resistance.

Ancient gut microbiomes, preserved in coprolites, dental calculus, and paleofeces, serve as paleogenomic "time capsules" that chronicle microbial evolution under pre-industrial diets, natural antimicrobial exposures, and negligible antibiotic pressure[9,10]. Notably, these communities exhibit compositional signatures distinct from modern microbiomes. One of these signatures is the dominance of *Segatella copri* (previously *Prevotella copri*)[11]. This taxon has been depleted in most of the modern gut microbiomes, possibly driven by processed foods, clinical antibiotics, and hygienic practices[12–14]. Given

the unique compositions of pre-industrial microbiota and the fast evolution rates of microbial strains, ancestral microbial ecosystems might harbor AMPs whose targets have no modern analogs[15]. Critically, these ancient AMPs were forged in biological conflicts against pathogens lacking contemporary resistance mechanisms, creating a potential evolutionary mismatch: their molecular targets may remain vulnerable in modern pathogenic strains[16]. By resurrecting AMPs from microbial lineages untainted in the industrial era, we tap into a pre-resistance arsenal with unparalleled therapeutic potential.

Here, we bridge evolutionary biology and antimicrobial discovery by leveraging the ancient gut microbiome for combating modern pathogenic microorganisms. Combining an AI-driven metagenomic mining tool (AMPLiT) with experimental validation, we identified huge amounts of *S. copri*-derived AMPs from 7 ancient human coprolites[17]. This finding is consistent with previous observations that bacteria from the *Prevotella* genus are prolific producers of AMPs[3,18]. These AMPs, characterized by unique sequential patterns absent in modern databases[19], demonstrated broad-spectrum efficacy against pathogens in vitro and promising effectiveness in murine infection models, comparable to conventional antibiotics like vancomycin and polymyxin B. Overall, this work contributes to a growing body of studies mining ancient samples for novel peptide antibiotics[3,4,18].

## Results

### AMPLiT: a framework for time-efficient training on resource-constrained devices

Computational approaches for accurately identifying AMPs in microbiome (e.g., ancient coprolite metagenomes) demand substantial computational resources for deployment and model training[1,18,20]. To overcome this, we first developed AMPLiT (Antimicrobial Peptide Lightweight Identification Tool), a computational framework optimized for portable hardware. By re-engineering the architecture of our previous AMPidentifier[5], we minimized reliance on computationally intensive convolutional kernels through three innovative computational blocks (Supplementary Figs. 1 and 2), without sacrificing predictive performances (Table 1).

The first block, inspired by paradigmatic multi-head attention mechanisms, significantly reduced training time by 43% (Model C vs. AMPidentifier). The second module, building on prior mathematical principles, synergized with the first to further streamline training costs ($3206 \pm 68$ s) and improved overall performance (Model E, AUPRC: $0.9222 \pm 0.0031$), compared to the comparison framework (Model A, AUPRC: $0.8697 \pm 0.0025$). The third module introduced dynamic feature refinement to suppress noise in peptide sequences, achieving a 36.8% reduction in training time and an AUPRC of $0.8927 \pm 0.0086$ (Model B) compared to the original AMPidentifier[5]. The integration of the 3 blocks into AMPLiT (Model G) reduced approximately 80% training time ($3200 \pm 53$ s), while matching its state-of-the-art accuracy (AUPRC: $0.9486 \pm 0.0003$). Remarkably, the performance was attained using consumer-grade hardware (Intel i7-

10875H CPU), eliminating dependence on high-performance computing infrastructure.

### Mining and in vitro validation of AMP candidates from ancient gut microbiomes

We next analyzed seven published ancient *Homo sapiens* coprolite (1000–2000 years of history) metagenomes[17] using this tool (Fig. 1a). To identify ancient AMPs candidates, we screened the seven published ancient *Homo sapiens* coprolite (1000–2000 years of history) metagenomes[17] using AMPLiT (Fig. 1a). Following decontamination to remove environmentally derived sequences, we extracted raw open reading frames (ORFs) across all samples (Fig. 1b). Applying a stringent prediction threshold (probability ≥ 0.9), AMPLiT deployable on standard hardware was used to identify putative AMP candidates.

Comparative analysis of physicochemical properties (isoelectric point, aromaticity, charge at pH 7.0, and hydrophobicity) between AI-predicted sequences and 8794 experimentally validated AMPs revealed alignment in feature patterns (Fig. 1c–f), confirming the tool's capacity to capture features of AMPs. Intriguingly, the AI-predicted candidates exhibited significantly lower molecular weights (median ~2 kDa vs. 2.8 kDa), which might be confounded by the vertebrate-derived AMPs with longer sequences in the reference database. To prioritize candidates with synthetic feasibility and ecological relevance, we implemented a multi-step filtration strategy: sequences were first filtered by ORF length (39–78 nucleotides, corresponding to 13–26 amino acids) and sample coverage (occurred in at least 5 stool samples), narrowing the list to 160 non-redundant candidates. Subsequent cytotoxicity prediction[21] removed sequences with potential host toxicity, resulting in 41 high-confidence candidates for experimental validation.

Among the 41 filtered candidates, 40 peptides were successfully synthesized for in vitro validation (Fig. 1g and Supplementary Table 1). Initial screening against model Gram-positive (*Bacillus subtilis* 23857, *Staphylococcus aureus* 6538) and Gram-negative (*Escherichia coli* 25404, *Pseudomonas aeruginosa* 27853) pathogens revealed that 36 peptides (90% of those synthesized) exhibited measurable growth inhibition at 100 μM (Fig. 1h), underscoring the precision of AMPLiT. Strikingly, around two-thirds of the predicted AMPs originated from *S. copri*, a keystone commensal highly enriched in ancient gut microbiomes (Supplementary Fig. 3a) but depleted at levels of prevalence and abundance among modern industrialized populations[12,13]. Approximately 50% of the *S. copri*-derived AMPs showed potent antimicrobial activities (with at least 50% inhibitory rate to at least one tested bacterial strain), suggesting *S. copri* may play a critical role in maintaining microbial equilibrium through AMP production, a function potentially eroded during human modernization.

### Genomic origins and conservation of ancestral antimicrobials in *S. copri*

To elucidate the genomic origins of AMPs from *S. copri*, we analyzed the loci encoding AMPs with potent activity in vitro (Fig. 1h), revealing

**Table 1 | The overall performance of different strategies for AMPLiT**

| Method | TP | TN | FP | FN | Sensitivity (%) | Specificity (%) | AUPRC |
|---|---|---|---|---|---|---|---|
| AMPidentifier 1.0[5] | 854 ± 16 | 38201 ± 7 | 62 ± 7 | 135 ± 16 | 86.2963 ± 1.6929 | 99.8380 ± 0.0204 | 0.9495 ± 0.0022 |
| A | 790 ± 70 | 38,041 ± 101 | 221 ± 101 | 199 ± 70 | 79.8653 ± 7.0930 | 99.4207 ± 0.2654 | 0.8697 ± 0.0025 |
| B | 854 ± 24 | 38,016 ± 95 | 246 ± 95 | 136 ± 24 | 86.2626 ± 2.4632 | 99.3562 ± 0.2485 | 0.8927 ± 0.0086 |
| C | 839 ± 11 | 38,094 ± 27 | 168 ± 27 | 151 ± 11 | 84.7475 ± 1.1546 | 99.5601 ± 0.0727 | 0.8987 ± 0.0111 |
| D | 884 ± 105 | 38,078 ± 185 | 185 ± 42 | 105 ± 11 | 89.3603 ± 1.2018 | 99.5165 ± 0.1098 | 0.9224 ± 0.0075 |
| E | 872 ± 24 | 38,103 ± 33 | 159 ± 33 | 117 ± 24 | 88.1145 ± 2.4692 | 99.5836 ± 0.8750 | 0.9222 ± 0.0031 |
| F | 884 ± 39 | 37,989 ± 120 | 274 ± 120 | 105 ± 39 | 89.3266 ± 3.9470 | 99.2839 ± 0.3148 | 0.9172 ± 0.0074 |
| G (AMPLiT) | 892 ± 9 | 38,143 ± 17 | 97 ± 9 | 120 ± 17 | 90.1347 ± 0.9488 | 99.6864 ± 0.0459 | 0.9486 ± 0.0003 |

TP, FN, TN, and FP listed above represent true-positive number, false-negative number, true-negative number, and false-positive number, respectively.

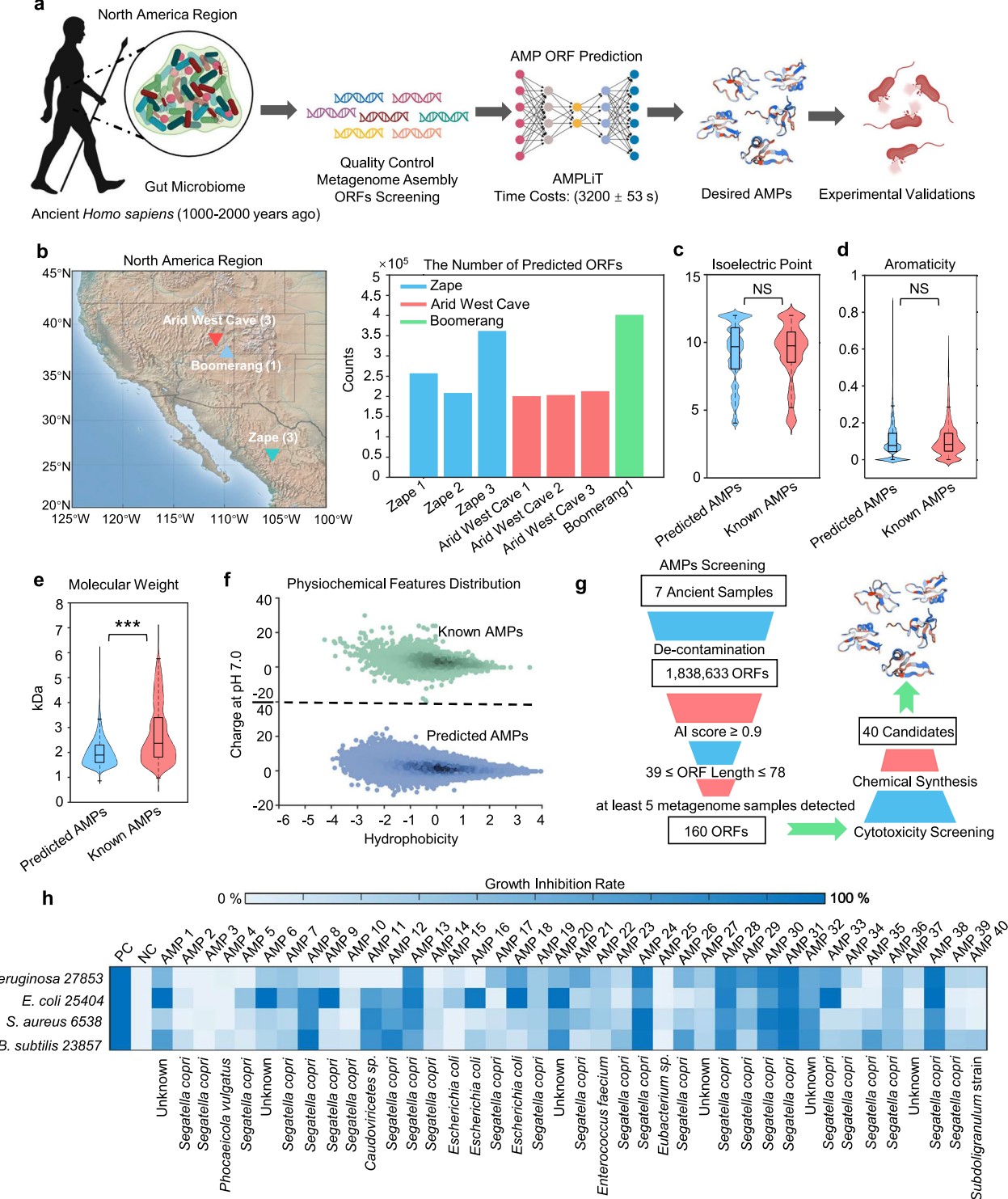

**Fig. 1 | The integrated mining strategy of AMPs from ancient biological samples. a** The bioinformatic approaches for ancient AMPs mining tasks. Gut microbiome sequences were assembled form ancient biological samples of human stool, and AMPs were mined from these data. Further screened by potential cytotoxicity, the selected candidates were experimentally validated in vitro and in vivo. **b** The prediction of AMP ORFs from seven well-processed palae-feces samples of *Homo sapiens* (1000−2000 years ago) with geographical locations. **c**−**f** The comparison of molecular features including isoelectric point, aromaticity percentage, molecular weight, charge at pH 7.0 and Hydrophobicity between previously reported AMPs ($n = 8794$) and Predicted AMPs ($n = 723,045$) from ancient samples. The statistical tests were conducted by 2-sided Welch T test with the adjustment of Cohen's d method ($p < 0.05$ and |Cohen's d | > 0.8, *, $p < 0.01$ and |Cohen's d | > 0.8, **,

$p < 0.001$ and |Cohen's d | > 0.8, ***). The 2-sided Welch T test *p*-value in molecular weight comparison is less than $1 \times 10^{-16}$. **g** Screen criteria of AMP candidates for further experimental validations. **h** The experimental validations for antimicrobial activities of the predicted 40 peptides from ancient biological samples towards *P. aeruginosa 27853*, *E. coli 25404*, *B. subtilis 23857*, and *S. aureus 6538* in vitro. PC and NC represent positive control and negative control, respectively. The NC represents 1× PBS solutions. The PC represents 100 μM kanamycin for groups of *E. coli 25404*, *B. subtilis 23857*, and *S. aureus 6538* or 100 μM polymyxin B for the group of *P. aeruginosa 27853*. BioRender was used during the construction of the Figure, with the issued License agreement number GG294KREHV (Created in BioRender. Chen, S. (2025) https://BioRender.com/7b397jf).

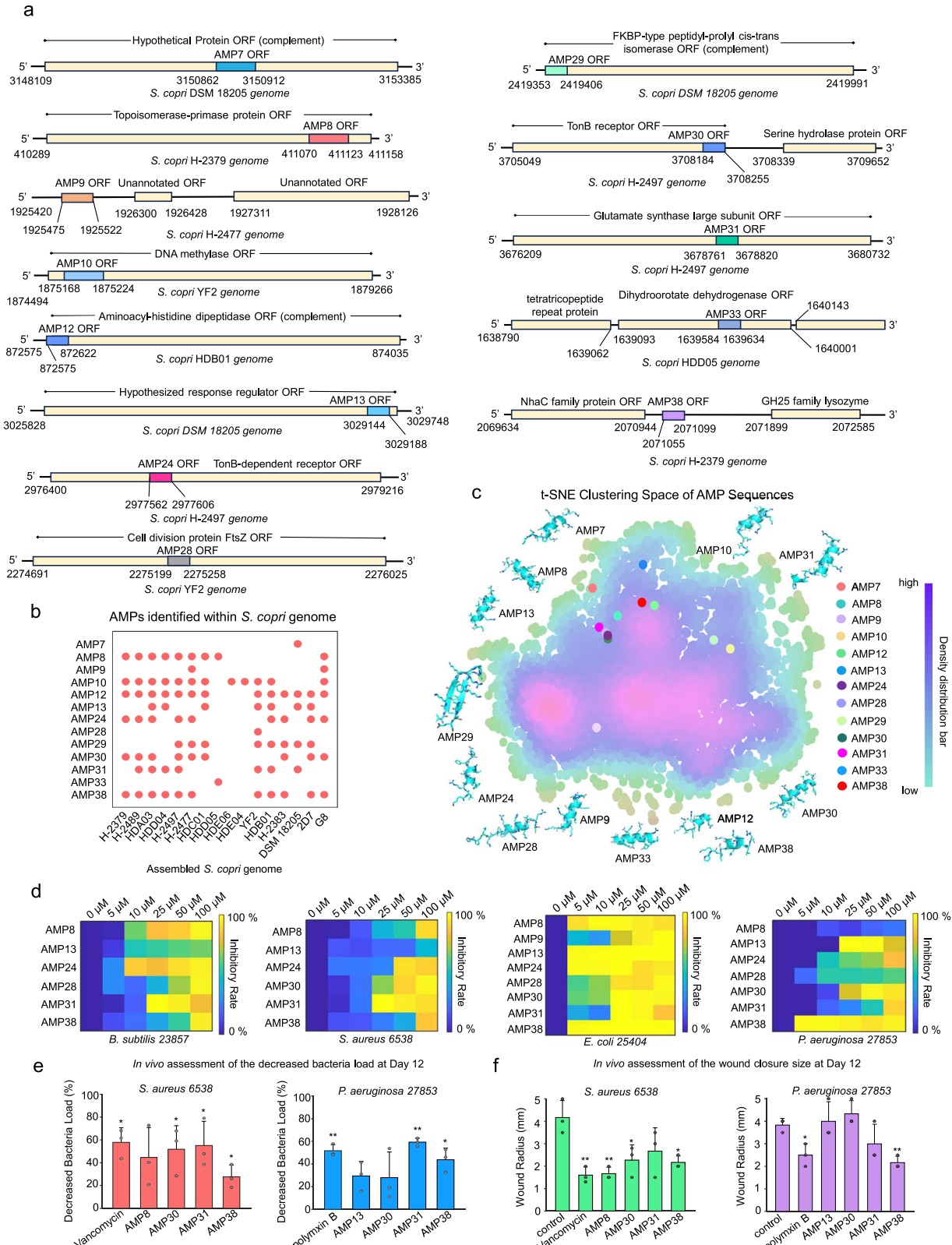

interesting evolutionary and functional insights (Fig. 2a). Notably, around 77% of those ORFs represented fragments of larger genes encoding functionally diverse proteins, including topoisomerase-primase domain-containing proteins, aminoacyl-histidine dipeptidase, cell division protein FtsZ, TonB receptors, glutamate synthase large subunits, and etc. (Fig. 2a and Supplementary Fig. 4). This pattern aligns with recent reports of resource-efficient evolutionary strategies in which microbes repurpose "trans-membrane" fragments or motifs of larger housekeeping genes to generate novel AMPs, a mechanism thought to minimize metabolic costs while expanding antimicrobial repertoires[8,9,18,22]. In contrast, AMP9, AMP24, and AMP38 ORFs were not embedded within any annotated functional genes, suggesting alternative origins such as noncoding regions or horizontal gene transfer.

**Fig. 2 | The identification of AMP ORFs from *S. copri* genomes reveal conservative and unique patterns in sequence compositions. a** The length and distribution of particular AMP ORFs on the assembled genomes of *S. copri* collected from rural populations. By default, the coding direction of the annotated genes follows the 5′ to 3′ direction. Those annotated with "complement" tags indicate the distribution along the reverse strand. **b** The existence of particular AMP ORFs screened on the currently available *S.copri* genomes in the public dataset. For the assembled metagenomic contigs containing these AMP ORFs from ancient prevalent gut microbiota *S. copri*, the conservations of these AMPs were further confirmed through alignment-based taxonomic binning against reference genomes available in NCBI. **c** t-SNE clustering methods on 8794 publicly known AMP sequences and 13 representative AMP sequences identified from *S. copri* in the 7 ancient stool metagenomic samples. The structures of AMPs were predicted by using Alphafold3[44]. **d** The antimicrobial efficacy of the representative AMPs from *S.* *copri* with diverse concentration gradients against *P. aeruginosa 27853*, *E. coli 25404*, *B. subtilis 23857*, and *S. aureus 6538*. **e** The in vivo assessment of the decreased bacteria loads (left panel: $p = 0.0153$, $p = 0.0979$, $p = 0.049$, $p = 0.0454$, $p = 0.0415$) (right panel: $p = 0.0316$, $p = 0.0543$, $p = 0.1649$, $p = 6.83 \times 10^{-5}$, $p = 0.015$) and **f** the wound closure radius by the representative AMPs (left panel: $p = 0.0152$, $p = 0.0193$, $p = 0.033$, $p = 0.1208$, $p = 0.0327$) (right panel: $p = 0.0248$, $p = 0.7769$, $p = 0.2739$, $p = 0.2322$, $p = 0.021$). The data were presented as mean ± SD, and statistical tests between the control group and experimental group were statistically conducted by a 2-sided Welch T test ($p < 0.05$, *, $p < 0.01$, **, $p < 0.001$, ***), with 3 independent replicates in each group involved. BioRender was used during the construction of the Figure, with the issued License agreement number GG294KREHV (Created in BioRender. Chen, S. (2025) https://BioRender.com/7b397jf).

Due to DNA degradation in the ancient samples, the precise binning for these ancient AMP ORFs into particular phylogenetic lineages and clades of *S. copri* is currently difficult. Nonetheless, multiple ancient ORFs were consistently identified across multiple publicly available *S. copri* genomes assembled from rural populations (Fig. 2b). Some ORFs encoding broad-spectrum antimicrobials (AMP8, AMP13, AMP31, AMP38) were even conservatively shared among at least six distinct *S. copri* strains, indicating striking conservation. Notably, more than 67% of the ORFs detected in *S. copri* were not shared with other species, suggesting exclusive distribution patterns. Furthermore, sequence space clustering revealed limited overlap between these ancient AMPs and currently known AMPs (Fig. 2c), highlighting their unique signatures. Together, these findings suggest that *S. copri* evolved specialized AMPs potentially through both gene fragment recycling and de novo innovation. The exclusive distributions of these peptides seemed to imply unique roles of *S. copri* in strain-specific colonization or niche interactions with host cells, a dynamic now obscured by its decline in modern microbiomes.

A dose-response assay further benchmarked the therapeutic potential of the top-performing *S. copri*-derived AMPs against multiple pathogens (Fig. 2d). For Gram-positive pathogens, six peptides (AMP8, AMP13, AMP24, AMP30, AMP31, and AMP38) effectively suppressed *B. subtilis* 23857 and *S. aureus* 6538 at 25–100 μM. For Gram-negative strains, eight candidates (AMP8, AMP9, AMP13, AMP24, AMP28, AMP30, AMP31, and AMP38) showed potent inhibitory efficacy against *E. coli* 25404 at 5–100 μM, while five peptides (AMP13, AMP24, AMP30, AMP31, and AMP38) revealed inhibitory efficacy against *P. aeruginosa* 27853 at 5–100 μM. Moreover, we observed a non-monotonic concentration-dependent antimicrobial effect for certain resurrected AMPs, wherein inhibitory activity decreased at higher concentrations. This phenomenon aligns with previously reported behavior of AMPs, which may undergo self-association or aggregation at elevated concentrations[7,23,24], thereby influencing their observed efficacy. The results of lead peptides were further tested by CFU plate spreading methods (Supplementary Fig. 3b, and Supplementary Table 2). Overall, these observations demonstrated broad-spectrum inhibitory efficacy by *S. copri*-derived AMPs, highlighting the functional decline of protective commensals in modern gut ecosystems.

### AMPs from commensal *S. copri* displayed low cytotoxicity risks
Given the historical symbiosis between *S. copri* and ancient humans, we hypothesized that these AMPs would exhibit low cytotoxicity-a trait evolutionarily optimized for coexistence with mammalian hosts. To assess this, we prioritized seven candidates (AMP8, AMP13, AMP24, AMP28, AMP30, AMP31, and AMP38) showing high growth inhibitory rate (≥70%) and broad-spectrum activities (≥2 strains) for further safety profiling. Hemolysis assays revealed minimal or undetectable red blood cell lysis across all tested concentrations (5–100 μM), with no candidate exceeding a 10% average hemolytic rate (Fig. 3a). Similarly, CCK-8 cytotoxicity assays using intestinal epithelial cells showed no significant viability reduction at concentrations 5–100 μM for most peptides (Fig. 3b–h), except for slight cytotoxicity by AMP30 at 100 μM and AMP38. Scanning electron microscopy (SEM) further demonstrated the peptides' mechanistic selectivity: all seven candidates disrupted the membrane integrity of pathogenic Gram-positive (*S. aureus*) and Gram-negative (*E. coli* and *P. aeruginosa*) pathogens (Fig. 3c). The discriminatory actions suggests that *S. copri* fine-tuned these AMPs to balance antimicrobial efficacy with biocompatibility, a dual functionality rarely achieved by synthetic peptides[25].

### In vivo effectiveness of AMPs from *S. copri*
Building on their efficacy against *S. aureus* 6538 and *P. aeruginosa* 27853 in vitro (Fig. 2d and Supplementary Fig. 3c), five AMPs (AMP8, AMP13, AMP30, AMP31, and AMP38) were advanced to in vivo test using murine wound infection models (Fig. 4a and Supplementary Fig. 5a). Mice infected with either pathogen were treated topically with AMPs (100 μM) at Days 1, 2, 3, 6, and 9, with vancomycin (100 μM) and polymyxin B (100 μM) serving as antibiotic controls. By day 12, multiple AMP treatment groups in combating gram-positive pathogens showed observable reductions in bacterial load (Fig. 4b) and accelerated wound closure (Fig. 4c and d). For *S. aureus*-infected wounds, AMP30, AMP31, and AMP38 showed antimicrobial efficacy comparable to vancomycin, achieving near-complete wound healing (Figs. 2e, f and 4c). Histological investigations of wound tissues using H&E and Masson staining showed severe immune cell infiltration in the negative control group, whereas experimental groups of AMP8, AMP30, and AMP38 showed lower immune cell infiltration similar to vancomycin treatment group (Fig. 4d and Supplementary Fig. 5b). For *P. aeruginosa*-infected wounds, AMP31 and AMP38 treatment groups showed improvement in antimicrobial efficacy (Fig. 2e). Improved wound healing and lower immune cell infiltration in AMP38 and polymyxin B-treated groups were also observed, indicating slightly improved tissue recovery (Figs. 2f and 4c). These results demonstrated the efficacy of *S. copri*-derived AMPs, especially in combating gram-positive pathogenic bacteria.

## Discussion
This study demonstrates that ancient gut microbiomes, preserved in coprolites, harbor AMPs with untapped therapeutic potential. By developing AMPLiT, a lightweight AI-driven computational pipeline, we successfully synthesized 40 high-confidence AMP candidates from seven ancient human coprolite metagenomes, 36 of which exhibited potent in vitro activity against multiple prevalent pathogens. Nearly two-thirds of active AMPs originate from *S. copri*, an evolutionarily significant gut commensal largely absent in modern industrialized populations. These AMPs displayed broad-spectrum antimicrobial efficacy, low cytotoxicity, and accelerated wound healing in murine models, rivaling clinical antibiotics like vancomycin and polymyxin B.

AMPLiT addresses a critical bottleneck in AMP discovery: the reliance on high-performance computing for large-pipeline training

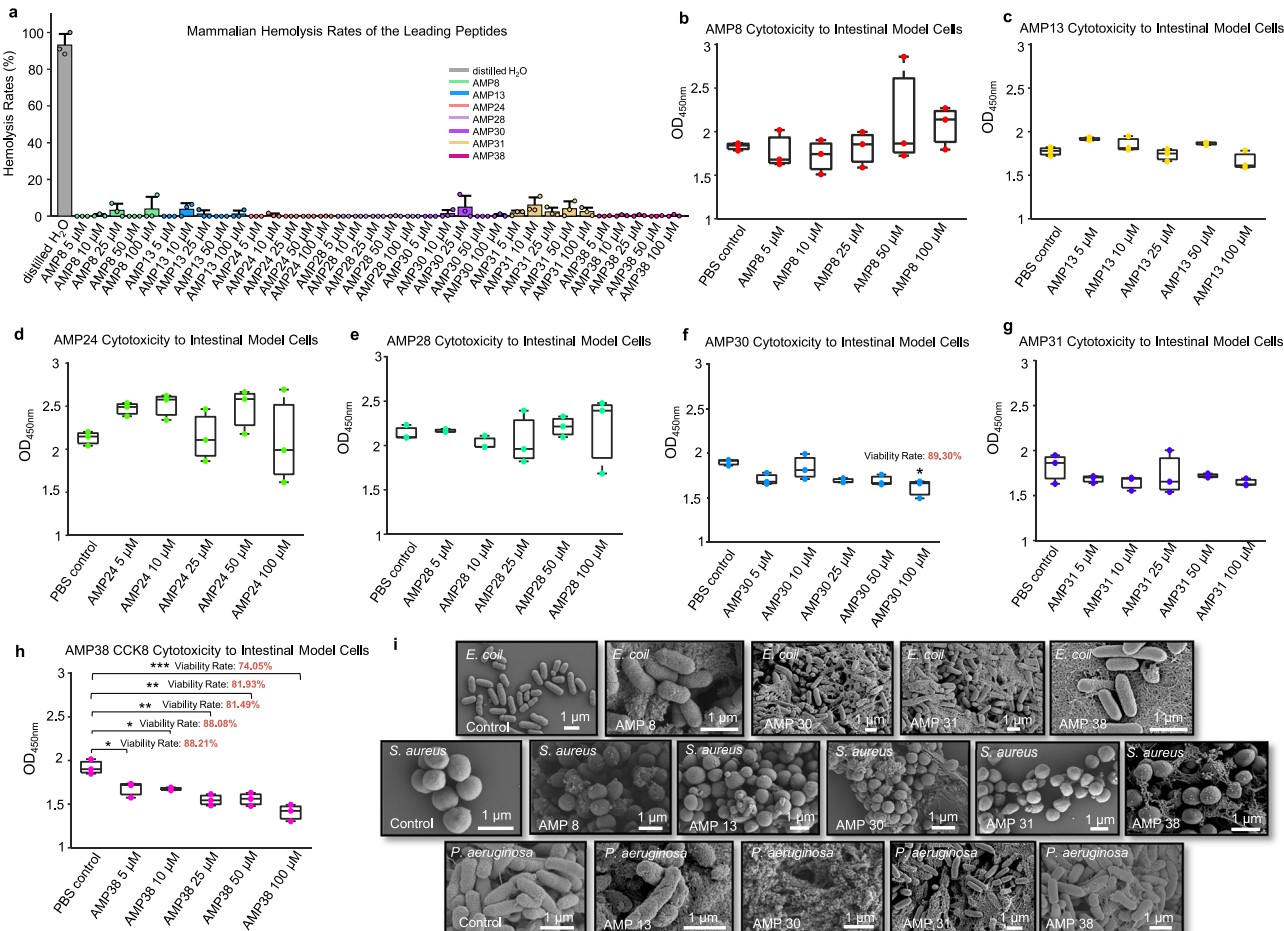

**Fig. 3 | The bio-safety and mechanistic assessment of the selected leading AMPs. a** The hemolytic cytotoxicity of the selected AMPs was conducted by co-culturing with 4% (v/v) murine blood cells at 37 °C, 2 h, with distilled water and normal saline solution as positive control and negative control, respectively. The hemolytic cytotoxicity was quantified by experimentally testing $OD_{450}$ with three replicates, and all groups deducted the absorbance values of the negative control group. Standard formula was subsequently used to calculate the hemolysis rates. The data were presented as mean ± SD and the statistical significance was conducted by 2-sided Welch T test ($p < 0.05$, *, $p < 0.01$, **, $p < 0.001$, ***). All the experimental groups were conducted with three replicates. **b–h** The CCK8 cytotoxicity in vitro assessment of the selected AMPs by co-culturing with Caco-2 cells mimicking in vivo small intestinal epitheliums, at 37 °C, 12 h. The cell viability of those experimental groups with significant differences (One-way ANOVA-test

($p$* ≤ 0.05, $p$** ≤ 0.01, $p$*** ≤ 0.001) combined with Post-hoc multiple comparisons) compared to the control group were calculated by the standard formula. The box plots show the median (central line), the 25th percentile and 75th percentile, and interquartile range. All experimental groups were conducted with three independent replicates. AMP8, $p$-value = 0.9999, 0.9972, 1.0000, 0.7962, 0.9283; AMP13, $p$-value = 0.1765, 0.7083, 0.9806, 0.5713, and, 0.3634; AMP24, $p$-value = 0.6955, 0.5948, 1.0000, 0.6851, 1.0000; AMP28, $p$-value = 1.0000, 0.9884, 0.9975, 0.9982, and, 0.9998; AMP30, $p$-value = 0.1054, 0.9312, 0.0732, 0.0756, and, 0.0106; AMP31, $p$-value = 0.8062, 0.5982, 0.9620, 0.9373, and, 0.5814; AMP38, $p$-value = 0.0173, 0.0163, 0.0006, 0.0008, and, 0.00003. **i** The typical SEM morphologies of the destructed bacteria membranes by the representative AMPs (100 μM) co-culturing with different bacteria (1 × 10$^8$ cfu/mL), at 37 °C, 24 h.

and accurate AMP mining from large-scale metagenomic data[1,4,5,9,18]. By parameter-efficient optimizations and designs in architectures of the framework, AMPLiT achieved robust predictive accuracy with an 80% reduction in training time, operable on consumer-grade hardware. This democratizes access to AMP mining in resource-limited paradigms, which are often not accessible because of computationally-expensive (multiple GPUs) or time-exhaustive (1–2 weeks training) barriers by using the conventional large pipelines[1,7,20,26]. Though machine learning algorithms showed themselves as cheap and accessible strategies, their insufficient performances and high false-positives limited further applications[7,9,18]. The AMPLiT showed promising performances in both accurately identifying functional AMPs from fragmented, low-biomass coprolite data and enabling time-efficient and computationally-balancing manners underscores its transformative potential for evolutionary microbiology and drug discovery.

Ancient microbial samples, including coprolite metagenomes, serve as genomic "time capsules"[10,17,27], preserving microbial communities unexposed to industrial-era antibiotics. These ancient microbiomes encode AMPs evolved under pre-industrial selective pressures, potentially targeting conserved bacterial components that may remain vulnerable in modern pathogens. Unlike modern antimicrobial agents[2,8,16,28], which face resistance mechanisms honed over decades[6,19,25,29,30], ancient peptides potentially circumvent this evolutionary arms race[4,9]. Our discovery of putative AMPs with relatively low homology to modern databases[19,25,29,30], highlights the coprolite microbiome as a valuable reservoir for novel antimicrobials. Though the decline of *S. copri* in Westernized populations has been widely observed[14,31,32], many of the AMP ORFs were also identified across multiple publicly available *S. copri* genomes assembled from rural populations, suggesting exclusive distributions of these ORFs in *S. copri*. They were embedded in diverse genomic contexts, including

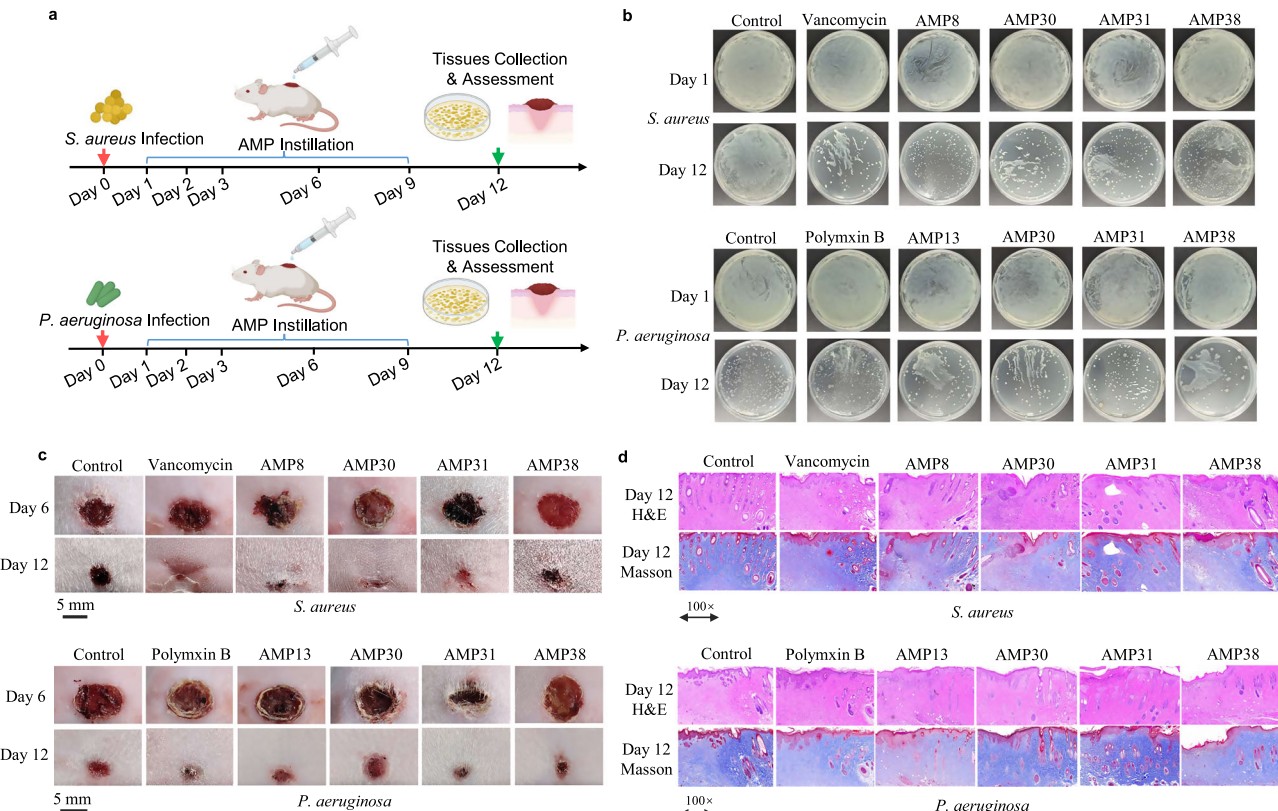

**Fig. 4 | In vivo efficacy of the representative AMPs from primitive commensal *S. copri* against *S. aureus* 6538 and *P. aeruginosa* 27853. a** The experimental design for assessing in vivo treatment effectiveness by AMPs from *S. copri*. **b** Experimental confirmation of pathogens infection load in control, vancomycin/polymxin B, and AMP8, AMP13, AMP30, AMP31, and AMP38 treatment groups by streaking on LB agar plates. **c** Representative in vivo wound healing results by negative control PBS, AMP groups, and positive control vancomycin or polymyxin B at the same dosage in vivo. **d** The representative histological results of mice wound tissues from groups of control, vancomycin/polymyxin B, AMP8, AMP13, AMP30, AMP31, and AMP38 generated using H&E and Masson staining protocols. BioRender was used during the construction of the Figure, with the issued License agreement number GG294KREHV (Created in BioRender. Chen, S. (2025) https://BioRender.com/7b397jf).

housekeeping gene fragments. This resource-efficient evolutionary strategy[4,6,9,18] repurposes trans-membrane domains of larger functional proteins into AMPs, suggesting *S. copri* optimized its antimicrobial arsenal for metabolic efficiency. The *S. copri*-derived AMP repertoire likely played a dual role in pathogen suppression and ancient host-microbe symbiosis. The decline of this keystone species in modern guts may contribute to dysbiosis and infection susceptibility[11,33], positioning its AMPs as both therapeutic agents and ecological restoration tools.

Despite these promising findings, there are several limitations. Notably, the relationship between genes and proteins is not fully congruent in certain cases, as evidenced by previous studies reporting antimicrobial activity in certain encrypted peptides[9,18], functional peptides encoded within genomic or transcriptomic sequences that are overlooked by conventional gene prediction algorithms. Our current approach for metagenome-based mining of AMPs, in rare edge cases, may ignore some encrypted proteins due to post-translational modifications or potential overlaps in the predicted ORFs, and further studies are needed to unlock the full potential of ancient microbiome-derived AMPs. Also, the transcriptional activity of the identified AMP ORFs needs to be further confirmed, and the exact ecological role of them in ancient gut microbiomes remains a subject for future exploration. More in silico and experimental efforts (e.g., proteomes, transcriptomics, and multi-omics approaches) are needed to elucidate the natural occurrence of these AMPs. The CFU counts were not quantitatively performed in the in vivo wound model due to practical challenges in reliably isolating and enumerating colonies from tissue homogenates. Although the alternative approach assessed via $OD_{600}$ of tissue supernatants and subsequent qualitative plating co-indicated reductions in bacterial burden, more experimental efforts are needed to overcome the limitations. A systematic in vitro dose-response assay comparison with clinical antibiotics like Vancomycin was beyond the scope of this discovery-phase study, but is also essential in future work. Furthermore, a systematic assessment of peptide efficacy against contemporary antimicrobial-resistant pathogens is imperative and constitutes a key goal for subsequent research. The assessment of their impact on the modern host commensal community and immune system should also be conducted to assess potential immunogenicity, which is essential for further safety evaluation. Additionally, the therapeutic potential of these peptides could be further enhanced by exploring stability improvements (e.g., introducing D-amino acids, cyclization) and synergistic combinations. Lastly, future efforts should also expand coprolite sampling across eras and cultures while exploring synergies between ancient AMPs and modern therapies, a critical step toward reclaiming the protective microbiota eradicated by industrialization.

Overall, this work explored "lost" AMPs from ancient microbiomes, offering a new perspective in antibiotic discovery. The identification of functional peptides from evolutionarily distinct microbiomes, coupled with the efficacy and biocompatibility of *S. copri*-derived AMPs, demonstrates that solutions to combating pathogenic microorganisms may lie in humanity's microbial past.

## Methods

### Biological data availability and metagenome assembly

The quality control steps follow the procedures of the previous study[17], and 7 metagenomic samples among 15 palaeofaeces samples that have been previously confirmed with low contaminations[17] were curated for the subsequent study. In details, 7 ancient human metagenomic stool samples (SRR12557704, SRR12557705, SRR12557706, SRR12557722, SRR12557711, SRR12557707, and SRR12557733), and 3 environmental metagenomic samples (SRR12557702, SRR12557703, and SRR12557732)[17] were collected from NCBI for AMPs mining. First, the reads simultaneously detected in stool samples and environmental samples were regarded as environmental contaminations and were removed. Then, human host DNA sequences were excluded by using KneadData v.0.6.1 and the *Homo sapiens* reference database (build hg19)[34]. Lastly, short reads fewer than 30 bps were also removed. The metagenome assembly was conducted by SPADes (v4.0.0)[35], with modes of "meta", and automatically optimized k-mer parameters.

### ORFs mining from ancient metagenomic samples

To avoid false positives, the assembled contigs contain "N" in sequences. Next, an in-house script named "ORF_hunter.py" created in our previous study[5] was used for mining raw ORFs (33-150 nt). The length cutoff range was set in accordance with the peptide length scope in the dataset used for model training and in consideration of downstream chemical synthesis applicability. The "ORF_hunter.py" functionally detects the ORFs within assembled contigs and converts them to the corresponding amino acid sequences, with 1,838,633 peptide sequences reserved.

### Model training and peptide prediction

The AMPLiT was created by using Python 3.9, with the original code publicly available at Github: https://github.com/ChenSizhe13893461199/AMPLiT. The data used for model training were curated from UniProt[36], and AMP databases[7,19,29,30,37], with procedures for constructing training dataset, validation dataset, and test dataset strictly consistent with our previous study[5]. Overall, 8859 AMPs and 79,675 non-AMPs with length less than 50 amino acids were reserved, with the training dataset (7065 AMPs and 36,339 non-AMPs), the validation dataset (804 AMPs and 5006 non-AMPs), and the test dataset (990 AMPs and 38,263 non-AMPs), respectively. As outlined in the previous study[5], the imbalanced ratios of AMPs/non-AMPs cases don't signify the imbalanced featural identification between AMPs/non-AMPs, as performances in sensitivity and specificity (Table 1) are balanced. The introduction of many more non-AMPs has been validated to be necessary, as insufficient coverage of non-AMPs (molecular features are diverse) in model training is risky in leading to false positives. Therefore, compared to the paradigm of the "focal loss" method, the binary cross-entropy strategy was selected and showed superior effectiveness in model training. The measures of decreasing model "over-fitting" dilemma are conformed to the methods used in our latest article[5]. Multiple tests on the performances of AMPLiT confirmed the optimal model structures and parameters (Fig. S2 and Table 1). The core model structure includes two nine-layer dense blocks, with filter kernel sizes of $1 \times 15$, $1 \times 75$, and $1 \times 15$, respectively. The initial learning rate was set to 0.003 using the Adam optimizer with default parameters. The dropout rate and dropout density were both set to be 0.2, with batch sizes of 512. Weight decay was set as $1 \times 10^{-6}$, with the growth rate of dense blocks as 16. The aforementioned hyperparameters were confirmed by model fine-tuning over the validation dataset and final performance assessment on the test dataset. Compared to AMPidentifier 1.0 embedded with computational block 2 (inspired by transformer frameworks), we additionally designed two new computational blocks to decrease computational burdens while retaining the robust performances. The first block, inspired by paradigmatic multi-head attention mechanisms, significantly reduced

training time by 43% (Model C vs. AMPidentifier). The second module, building on prior mathematical principles, synergized with the first to further streamline training costs ($3206 \pm 68$ s) and improved overall performance (Model E, AUPRC: $0.9222 \pm 0.0031$), compared to the comparison framework (Model A, AUPRC: $0.8697 \pm 0.0025$). The third module introduced dynamic feature refinement to suppress noise in peptide sequences, achieving a 36.8% reduction in training time and an AUPRC of $0.8927 \pm 0.0086$ (Model B) compared to the original AMPidentifier[5]. The integration of these 3 blocks into AMPLiT (Model G) achieved an approximately 80% reduction in training time ($3200 \pm 53$ s), while matching its state-of-the-art accuracy (AUPRC: $0.9486 \pm 0.0003$). Remarkably, this performance was attained using consumer-grade hardware (Intel i7-10875H CPU), eliminating dependence on high-performance computing infrastructure. AMPLiT generated parallel performances to AMPidentifier 1.0, with overall fitting parameters and time costs decreased by approximately 56 and 80%, respectively. All tests were conducted on standard laptops (Intel i7-10875H CPU), demonstrating reasonable resource requirements. For peptide descriptors, the methods of using one-hot code and physicochemical descriptors were the same as the methods outlined in our previously published article[5]. The one-hot matrices contained $n_{max} \times 20$ elements (where $n_{max} = 50$, representing the maximum amino acid numbers in peptide sequences) to represent sequence compositions. The protein physicochemical descriptors were calculated by Propy3 packages (https://propy3.readthedocs.io/en/latest/UserGuide.html), which generated a mathematical vector containing 1547 unique features of a given protein sequence. To represent the intrinsic relationships between amino acids within a particular peptide, the Word2Vec language framework was trained on short peptides (less than 50 amino acids) curated from the UniProt database[36] by setting the "subcellular location" filter. All the same cases shared with the validation and test datasets were excluded. By extensive tests, the method of skip-gram and "hierarchical softmax" were selected, with optimal parameters of context window, vector size for each amino acid, and minimum word count confirmed as 50, 100 and 1, respectively. Overall, the model input of AMPs/non-AMPs was simultaneously represented by matrices of one-hot codes, physicochemical descriptors and the aforementioned Word2Vec matrices, respectively. The validities of all these descriptors were tested through the performance evaluations.

By setting the threshold of AI score as 0.9, the AMPLiT was used for identifying potential AMPs among the 1,838,633 raw peptide sequences. Due to the challenges associated with chemical synthesis, only the ORF length between 39 and 78 base pairs were reserved, and this range is also consistent with most of the AMPs in databases[7,19,25,29,30]. Next, we reserved the sequences with ORFs detected in contigs of at least 5 ancient stool metagenomic samples, generating 160 non-redundant potential sequences. Then, the 160 sequences were further filtered by cytotoxicity prediction[21]. Finally, 41 peptides (40 peptides successfully synthesized) were reserved for experimental validations. For particular assembled metagenomic contigs containing these AMP ORFs, they were putatively assigned to the ancient prevalent gut microbiota *S. copri*, through alignment-based taxonomic binning against reference genomes available in NCBI. Although the inherent erosion of ancient microbial DNA precluded full genomic reconstruction and limited resolution to the species level, the high sequencing coverage of the corresponding contigs supports their assembly reliability and preservation quality. These contigs exhibited strikingly high identity to modern *S. copri* reference genomes without tracing to other species, confirming the phylogenetic origin of the *S. copri* clade.

### Model prediction evaluations

The AI deep learning model performances were mathematically evaluated by indicators of Sensitivity, Specificity, and AUPRC with

formulas shown below:

$$\text{Sensitivity} = \frac{\text{True positive number}}{\text{True positive number} + \text{False negative number}} \quad (1)$$

$$\text{Specificity} = \frac{\text{True negative number}}{\text{True negative number} + \text{False positive number}} \quad (2)$$

The AUPRC is the indicator quantifies the area under the precision-recall curve (PRC). PRC plots Precision against Recall. The formula of Precision and Recall are shown below:

$$\text{Precision} = \frac{\text{True positive number}}{\text{True positive number} + \text{False positive number}} \quad (3)$$

$$\text{Recall} = \frac{\text{True positive number}}{\text{True positive number} + \text{False negative number}} \quad (4)$$

### Chemical synthesis of the selected peptides

There are 40 peptides successfully synthesized by Gill Biotech's solid-phase approach[38], with one peptide excluded because of failure in chemical synthesis. The method began with resin functionalization, where the resin was treated with a 20% piperidine solution for 30 min to remove the N-terminal Fmoc group. The deprotection step was confirmed by using a colorimetric reagent that turned color of blue. Next, by adding a coupling reagent, a base, and Fmoc-X (any of the 20 standard amino acids) -OH, the reaction was initiated for an hour. The reaction completion was determined when the reagent showed no color change. The deprotection step was then repeated: the resin was again treated with 20% piperidine solutions for 30 min to remove the newly added Fmoc group. The effectiveness of the step was also confirmed by using the colorimetric detection reagent. Likewise, the aforementioned reactions were repeated until the ultimate peptide sequences were generated. Finally, the resin was carefully dried and cut to yield the crude peptide powder. For quality control, high-performance liquid chromatography and mass spectrometry were applied to determine the compositions and confirmed the molecular purities 95% at least.

### In vitro validations of antimicrobial activities

The antimicrobial activities of the selected peptides against *B. subtilis* 23857 (ATCC 23857), *S. aureus* 6538 (ATCC 6538), *E. coli* 25404 (ATCC 25404), and *P. aeruginosa* 27853 (ATCC 27853) were quantified according to the broth microdilution guidelines from NCCLS[39] and previously published article[1,5,9,20]. Specifically, the single colonies of the aforementioned bacteria were cultured in liquid Luria Broth (LB) medium to the exponential stage ($OD_{600}$ = 0.6) at 37 °C, 250 rpm. Next, the medium was diluted to a density of approximately $5 \times 10^5$ cfu/mL. The 100 μL aliquots were further transferred into a 96-well microtiter plate containing 100 μL AMP solutions (experimental groups), 100 μL kanamycin or polymyxin B solutions (positive control groups), and 100 μL deionized water (negative control groups). Notably, for tests against *B. subtilis* 23857, *E. coli* 25404, and *S. aureus* 6538, kanamycin was used as positive control groups. For tests against *P. aeruginosa* 27853, polymyxin B was used as positive control groups. The initial tests were conducted by using the final concentrations of 100 μM for both AMP experimental groups and positive control groups. The plates were incubated at 37 °C for 12 h, after which the $OD_{600}$ absorbance was determined by a microplate reader. All measurements were independently repeated 3 times. Subsequently, those candidate peptides from *S. copri* showing potent antimicrobial activities were further selected for antimicrobial tests in vitro under diverse concentration gradients of 0, 5, 10, 25, 50, and 100 μM, with the same experimental procedures and principles aforementioned.

### Hemolysis rates against red blood cells

The hemolytic tests were conducted according to experimental protocols applied in previously published articles[1,5,9]. The blood cells from healthy male rabbits (2–2.5 kg) were collected, and were centrifugated at 3000 rpm, 8 min for separating the plasma. Next, the upper plasma layer was aspirated and PBS solutions were added to wash the remaining red blood cells for 3 times, centrifugated at 3000 rpm for 10 min each time. Then, the remaining red blood cells were suspended to prepare 4% (v/v) solutions in 15 mL Eppendorf tube, and centrifugated at $1000 \times g$ for 10 min to obtain cell pellets, with supernatant discarded. The particular AMPs were dissolved in saline and cultivated with red blood cell pellets at final concentrations of 5 μM, 10 μM, 25 μM, 50 μM and 100 μM, respectively. Red blood cell pellets cultivated in 1 mL of saline and 1 mL of distilled water were regarded as a negative control and a positive control, respectively. All experimental groups were incubated at 37 °C for 2 h with three replicates for each group. Finally, all experimental groups were centrifuged at $1000 \times g$ for 10 min, collecting supernatants into a 96-well plate (100 μL each well) to measure $OD_{540}$ absorbance at 540 nm for confirming hemolysis rates. The formula for calculating the hemolysis rate is as follows:

$$\text{Hemolysis}(\%) = \left( \frac{OD_{\text{Sample}} - OD_{\text{Negativecontrol}}}{OD_{\text{Positivecontrol}} - OD_{\text{Negativecontrol}}} \right) \times 100\% \quad (5)$$

### Cytotoxicity against mammalian cells

Commercialized Caco-2 intestine model cells, widely used for mimicking the in vivo small intestinal epitheliums[40–43], were cultivated in T25-bottle containing 4 mL DMEM (20% Fetal Bovine Serum, and 1% Penicillin/Streptomycin) at 37 °C, 5% $CO_2$ atmosphere for 5 days. Subsequently, the Caco-2 cells were washed with PBS solutions for 3 times (20–30 s each time), followed by adding of 1 mL pancreatic enzymes for 3–5 min at 37 °C. Then, the suspended cells were centrifuged at $1000 \times g$ for 5 min. The cell pellet was resuspended in the DMEM (20% Fetal Bovine Serum, and 1% Penicillin/Streptomycin) to approximately $1 \times 10^4$ cells/mL and cultivated in 96-well plates (100 μL) for 36 h. Next, the cultivated solutions were added with 0.5% PBS, or 100 μM particular AMPs for 12 h. Finally, 10 μL of CCK-8 reagents (BS350E, purchased from Biosharp Life Sciences, CHN) were added to each well for 4 h at 37 °C. Each experimental group contain 3 replicates. The $OD_{450}$ of each well was measured after the incubation, and the cell survival rates were calculated based on these results.

### Scanning electron microscopy observation

The destruction effectiveness of particular AMPs to bacterial membranes was confirmed via using SEM (Thermofisher Helios 5 CX). The *S. aureus* 6538 (ATCC 6538), *E. coli* 25404 (ATCC 25404), and *P. aeruginosa* 27853 (ATCC 27853) were all cultivated in LB medium to the exponential phase at 37 °C, and suspended to the load of $1 \times 10^8$ cfu/mL, co-culturing with particular 100 μM AMP solutions at 37 °C for 24 h. Next, with 0.5% glutaraldehyde processed for 2 h, 10 μL liquid sample was added onto copper wafers. Finally, the pre-processed samples were sent to the SEM center at Kunming Medical University for generating SEM images.

### In vivo treatment using AMPs

All experiments using animals were consistent with the ethical policies, and the involved experimental protocols were approved by the Laboratory Animal Welfare & Ethics Committee at Health Science Center of Kunming Medical University (New Permit Number: 2025DF007). At Day 0, a total of 108 6–8 weeks healthy female Sprague-Dawley (SD) rats (190–220 g, purchased from Beijing Huafu-kang Biological Technology Co., Ltd.) were fully anesthetized and shaved on the back to establish infection models. Only female rats were used in this study to minimize sex-related variability in wound

healing and immune response. A full-thickness wound with 10 mm in circular diameters was created by using a hole puncher. Then, all mice were randomly separated into two groups, and the mice wound of either group were further dropped with suspensions of either *S. aureus* ATCC: 6538 or *P. aeruginosa* ATCC: 27853 ($1 \times 10^7$ cfu/mL, 100 μL), respectively. Following the 24 h infection, the mice infected by the same pathogens (*S. aureus* ATCC: 6538 or by *P. aeruginosa* ATCC: 27853) were randomly assigned again to one of six groups, respectively. The six groups contain: (1) PBS (negative control), (2) vancomycin (100 μM, for group infected by *S. aureus*) or polymyxin B (100 μM, for group infected by *P. aeruginosa*) (positive control), (3) AMP8 (100 μM, for group infected by *S. aureus*) or AMP13 (100 μM, for group infected by *P. aeruginosa*), (4) AMP 30 (100 μM), (5) AMP31 (100 μM), (6) AMP38 (100 μM). Following the steps, the mice wounds were administered with 100 μL of PBS, AMP8 or AMP13, AMP30, AMP31, AMP38, and vancomycin or polymyxin B on Days 1, 2, 3, 6, and 9, respectively. Three mice in each group were euthanized on Days 1, 6, and 12, with wound granulation tissues split into two sections for H&E/Masson staining and CFU load observations, respectively. The tissue portion for CFU load observations were firstly immersed with 2 mL PBS solutions for 2 h and then spread onto an LB agar plate at 37 °C for 14 h before assessment. In addition to the CFU load observations, the supernatant solutions immersed with the infected tissues were centrifuged at $1000 \times g$ for 5 min, and the OD600 values of the supernatant solutions were quantified to further assess and calculate the decreased bacteria load.

## Statistics and reproducibility

No statistical method was used to predetermine sample size. The exact choices of ancient metagenomic samples ($n = 7$) was determined based on the availability of published, pre-determined low-contamination coprolite datasets[17]. The data used for model training were curated from UniProt[36], and AMP databases[7,19,29,30,37], with procedures for constructing training dataset, validation dataset, and test dataset strictly consistent with our previous study[5]. Overall, 8859 AMPs and 79,675 non-AMPs with length less than 50 amino acids were reserved, with the training dataset (7065 AMPs and 36,339 non-AMPs), the validation dataset (804 AMPs and 5006 non-AMPs), and the test dataset (990 AMPs and 38,263 non-AMPs), respectively. No experimental data were excluded from the analyses during the computational and experimental process. The experiments were not randomized. The study did not include any interventions and thus the conventional blinding was not applicable to this study.

## Ethic statement

This research complies with the relevant ethical regulations. Experiments using animals were consistent with the ethical policies, and the involved experimental protocols were approved by the Laboratory Animal Welfare & Ethics Committee at Health Science Center of Kunming Medical University (New Permit Number: 2025DF007).

## Reporting summary

Further information on research design is available in the Nature Portfolio Reporting Summary linked to this article.

## Data availability

All metagenomic datasets used in this study are publicly accessible and can be freely used by readers without restrictions. In details, 7 ancient human metagenomic stool samples (SRR12557704, SRR12557705, SRR12557706, SRR12557722, SRR12557711, SRR12557707, and SRR12557733), and 3 environmental metagenomic samples (SRR12557702, SRR12557703, SRR12557732)[17] were downloaded from NCBI for AMPs mining. The computational and experimental data generated in this study are provided in the Source Data file. Source data are provided with this paper.

## Code availability

All code to reproduce data analyses is provided in the Supplementary Information. The original codes of AMPLiT for reproducing main and supplementary analyses were publicly available at https://github.com/ChenSizhe13893461199/AMPLiT (https://doi.org/10.5281/zenodo.17949815).

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

## Acknowledgements

Authors affiliated with MagIC are partially supported by InnoHK, the Government of Hong Kong, Special Administrative Region of the People's Republic of China. S.C.N. has received research funding from the Research Grants Council-Research Impact Fund (RGC-RIF, grant no. R4030-22), New Cornerstone Science Foundation (grant no. NCI202346), InnoHK, The Government of Hong Kong, Special Administrative Region of the People's Republic of China and the Leona M. and Harry B. Helmsley Charitable Trust (grant no. 2017PG-IBD003). This work was also supported by the National Key R&D Program of China (2024YFA1308300), National Natural Science Foundation of China (82460112), Applied Basic Research Projects of Yunnan Province, China (202201AW070019), "Xingdian Talents" Support Project of Yunnan Province (RLQB20220006), Yunnan Provincial Key Research and Development Plan (202403AC100020), The Innovative Team of Yunnan Province (202305AS350019).

## Author contributions

S.C., Y.Y., and Q.S. designed the research, analyzed the data, wrote the manuscript, and prepared the figures and tables. S.C. and Y.Y. conducted the experiment and analysis. Y.W. assisted with the data collection and analysis. S.C. and Q.S. designed the entire experimental process. Y.W., Y.P., H.M.T., Z.J., Y.M., L.S., X.Y., X.S., O.D., E.B.C., and F.K.L.C. provided comments and revisions on the manuscript. Y.S., Q.S., and S.C.N. supervised the research, coordinated the production of the data, and supervised the writing of the manuscript. S.C. and Y.Y. finalized the manuscript with input from all the authors. All the authors read and approved the final manuscript.

## Competing interests

S.C.N. has served as an advisory board member for Pfizer, Ferring, Janssen and Abbvie and received honoraria as a speaker for Ferring, Tillotts, Menarini, Janssen, Abbvie and Takeda; has received research grants through her affiliated institutions from Olympus, Ferring and Abbvie; is a founder member, non-executive director, non-executive scientific advisor and shareholder of GenieBiome Ltd which is non-remunerative; is a shareholder of MicroSigX Diagnostic Holding Limited; is a founder member, non-executive Board Director, and non-executive scientific advisor of MicroSigX Biotech Diagnostic Limited, which is non-remunerative; and receives patent royalties through her affiliated institutions. F.K.L.C. serves as the Principal Investigator for the Fecal Microbiota Transplantation Service under the Hospital Authority (HA). He is a Board Director of EHealth Plus Digital Technology Ltd., an HA-owned subsidiary driving the eHealth+ program to transform the Electronic Health Record Sharing System into a comprehensive digital healthcare platform and advance other IT initiatives within the eHealth ecosystem. Additionally, he is a Board Director of CUHK Medical Services Limited. F.K.L.C. is a shareholder of GenieBiome Holdings Limited and the co-founder, non-executive Board Chairman, and non-executive Scientific Advisor of its wholly owned subsidiary, GenieBiome Ltd. Similarly, he is a shareholder of MicroSigX Diagnostic Holding Limited and the co-founder, non-executive Board Chairman, and non-executive Scientific Advisor of its wholly owned subsidiary, MicroSigX Biotech Diagnostic Limited. He also serves as a Director of the Hong Kong Investment Corporation Limited and a member of the Steering Committee for the RAISe+ Scheme under the Innovation and Technology Commission. Furthermore, he is the Co-Director of the Microbiota I-Center (MagIC) Ltd. FKLC receives advisory fees and speaker honoraria from

AstraZeneca and Comvita New Zealand Limited, as well as patent royalties through affiliated institutions for microbiome-related applications. F.K.L.C., S.C.N., and H.M.T. are named inventors of patent applications held by the CUHK and MagIC that cover the therapeutic and diagnostic use of the microbiome. The remaining authors declare no competing interests.

## Additional information

[1]Microbiota I-Center (MagIC), Hong Kong SAR, China. [2]Department of Medicine and Therapeutics, State Key Laboratory of Digestive Diseases, Li Ka Shing Institute of Health Sciences, The Chinese University of Hong Kong, Hong Kong SAR, China. [3]Department of Geriatrics, Yunnan Geriatric Medical Center, The First Affiliated Hospital of Kunming Medical University, Kunming, Yunnan, China. [4]JC School of Public Health and Primary Care, Faculty of Medicine, The Chinese University of Hong Kong, Hong Kong SAR, China. [5]Li Ka Shing Institute of Health Sciences, Faculty of Medicine, The Chinese University of Hong Kong, Hong Kong SAR, China. [6]National Key Laboratory of Veterinary Public Health and Safety, Key Laboratory for Prevention and Control of Avian Influenza and Other Major Poultry Diseases, Ministry of Agriculture and Rural Affairs, College of Veterinary Medicine, China Agricultural University, Beijing, China. [7]School of Life Sciences, Gwangju Institute of Science and Technology, Gwangju, Republic of Korea. [8]School of civil and environmental engineering, Nanyang Technological University, Singapore, Singapore. [9]Singapore Phenome Center, Lee Kong Chian School of Medicine, Nanyang Technological University, Singapore, Singapore. [10]Department of Medicine, Section of Gastroenterology, Hepatology and Nutrition, University of Chicago, Chicago, IL, USA. [11]Centre for Gut Microbiota Research, The Chinese University of Hong Kong, Hong Kong SAR, China. [12]The D.H. Chen Foundation Hub of Advanced Technology for Child Health (HATCH), The Chinese University of Hong Kong, Hong Kong SAR, China. [13]Li Ka Shing Institute of Health Sciences, State Key Laboratory of Digestive Disease, Institute of Digestive Disease, The Chinese University of Hong Kong, Hong Kong SAR, China. [14]New Cornerstone Science Laboratory, The Chinese University of Hong Kong, Hong Kong SAR, China. [15]These authors contributed equally: Sizhe Chen, Yue Yuan. ✉e-mail: sunyang_doctor@vip.sina.com; siewchienng@cuhk.edu.hk; qisu@cuhk.edu.hk

