## [Transparent Peer Review File · Nature Communications]

Identification of antimicrobial peptides from ancient gut microbiomes

Corresponding Author: Professor Qi Su

Version 0:

Reviewer comments:

Reviewer #1

(Remarks to the Author)

Reviewer comments:

The manuscript presents a compelling study that utilizes portable hardware-compatible AI (AMPLiT) to mine novel antimicrobial peptides (AMPs) from ancient human coprolite metagenomes, predominantly derived from *Segatella copri*. The identification and subsequent validation of active AMPs against both Gram-positive and Gram-negative pathogens through *in vitro* and *in vivo* assays is commendable. The evolutionary rationale for exploring ancient microbiomes as a strategy to overcome modern resistance mechanisms is particularly insightful. However, it is essential to clarify that the identified AMPs are not traditional secreted bacterial peptides or host-derived AMPs, but rather fragments of various functional bacterial proteins. The rationale behind this approach requires further explanation. Additionally, the study did not include antimicrobial-resistant (AMR) strains in its analyses; consequently, the discovered AMPs did not demonstrate potent activity against AMR pathogens. This limitation should be explicitly acknowledged. To enhance the manuscript's rigor and impact, the authors should benchmark AMPLiT against existing AMP discovery methods to contextualize its performance and advantages. While the manuscript is generally well-structured, several key points—such as the methodology, rationale, and experimental scope—would benefit from clarification and expansion. Furthermore, the figure legends are currently insufficiently detailed and difficult to read; improving their clarity and comprehensiveness would greatly aid reader understanding.

Minor comments:

1. While the *in vitro* cytotoxicity results are presented, the potential for these evolutionarily distant AMPs to trigger heightened immunogenicity in a modern mammalian host (e.g., humans) warrants serious consideration. If feasible, preliminary cytokine profiling using human immune cells *in vitro* would significantly strengthen the safety profile assessment. At minimum, this must be highlighted as a key consideration for future development.
2. The paper focuses on pathogen suppression but ignores a key aspect: potential collateral damage to the host's beneficial microbiota. The antimicrobial peptides (AMPs) discovered by the authors are broad-spectrum, and their impact on modern commensal bacteria community is unclear.
3. Several references appear to be incorrectly cited, for example in Line 104, reference 19 is not relevant to toxicity prediction. In Line 123, references 7 and 17 do not seem to support the author's point of view. Please verify and replace with an appropriate citation.
4. More details could be added on the development of AMPLiT, such as to specify the transformer architecture used.
5. Did the authors predict AMPs from the metagenome assembled contigs or MAGs? And in Fig 1h and Line 112, how did the authors assign hosts to these AMPs? Please add this detail in the methods.
6. Line 241, the instability index is not shown in the figure.
7. Figure 2d, why does AMP31 or other AMPs have an inhibition rate of 100% at a concentration of 25 mM, but the inhibition rate decreases at higher concentrations?
8. The *in vivo* results are presented qualitatively, such as representative images. The lack of quantification severely limits interpretation and statistical validation.

(Remarks on code availability)

Reviewer #2

(Remarks to the Author)

This manuscript from Chen, Yuan, et al. presents several AMPs extracted from ancient gut microbiome data. They validate several candidates *in vitro*, with a high hit rate (albeit by using a high threshold of 100 μ M to define positive, see #3 below) and test the top candidate *in vivo* (but see #6 below).

Overall, I find that the work could be very interesting but several details are not clear enough to make a full assessment and some claims should be toned down.

MAJOR COMMENTS

1. Could the authors provide the codes in a more organized fashion? The authors mention that "All code to reproduce data analyses in this Brief Communication is provided in the Supplementary Information", but there is still a bit of a need to follow links to previous papers and repositories and nowhere is everything put together. At least a single document with instructions would be very helpful. In particular, I failed to find the code that produces produces the alignments shown in Fig 2A.
2. The argument of computational efficiency is not very convincing. In all these pipelines, the bottleneck is—by far—the assembly and not the AMP prediction step. For example, I can run the MACREL tool (which the authors indirectly cite as support for their claim as it is the tool behind [17]) on thousands of peptides per second (single CPU, using a laptop that was modern in 2021).
3. A threshold of 100 μ M is high for AMP determination. For the ones that the authors follow up on, they do use lower thresholds, but it would be good to either see the data for all AMPs or at least mention the high threshold already in the abstract (when the authors mention the 90% hit rate).
4. The lower molecular weight of the peptides here compared to the previously reported ones could be due to differences in origin species. Several peptides in the databases are from vertebrate origin, which which may influence the comparison. I am also do not find the inference that this may indicate potential membrane-penetration activity (L100) very convincing. Most AMPs are anyway membrane-disrupting, so is the argument here that the ancient ones are even more likely to be active in this way? In any case, the relationship with low microbial weight is not a strong one.
5. Also, if I understand the functionality of ORF_finder.py correctly (I'm reading the version here https://github.com/ChenSizhe13893461199/Fast-AMPs-Discovery-Projects/blob/2e1147625a96a6133875c309e5418882b9663d44/ORF_hunter.py which may not be the right version, but see #1), the authors actually consider the shortest possible ORFs even inside another longer ORF! It

is such an odd choice that I would tend to consider it a bug in the code. Also, if there is a out-of-frame START, it will reset the scanning to consider a different frame.

This also affects any claims of novelty. For example, taking just their first sequence (KERKKYGLKAARRAPQFSKR), this matches a longer peptide in AMPSphere: https://ampsphere.big-data-biology.org/amp?accession=AMP10.387_945 but given the way that ORFs are called here, only the shorter fragment is considered.

6. The claims related to the distribution of the *S. copri* AMPs in different strains (L130) are potentially very interesting, but it is not clear where the strain genomes are coming from. If the authors are using modern genomes, then this partly disproves the value of using ancient data (although it is very interesting from an evolutionary perspective to see that these genes have been conserved). In any case, the authors should provide more details on the genomes used.

7. The in vivo results are potentially very important, but there is a lack of quantification. The data is shown as an extended figure and it is, frankly, hard to be perform all the necessary comparisons by eye (and only a single replicate of each condition is shown). The authors should present the CFU counts as well and quantify the wound sizes in a systematic way.

MINOR COMMENTS

8. Finding *S. copri* as a high producer of AMPs is consistent with previous literature, see DOI 10.1016/j.cell.2024.05.013 or DOI 10.1016/j.cell.2024.07.027 (which use the name *Prevotella copri*).

9. Figs 1c-e: I would ask that the authors use box and dot plots instead of the dynamite ones which do not show the distribution of data well.

10. L113: what is the criterion behind calling some AMPs "highly potent"?

(Remarks on code availability)

See the remark on the main review: the code is only partially available or, at least it was not clear to me how to access all of it. Some important questions are not clear from the Methods/Figure legends and without the code, it was impossible to fully assess the paper.

Reviewer #3

(Remarks to the Author)

This paper by Chen et al. introduces AMPLIT, a deep-learning pipeline that can be run on consumer-grade CPUs to mine metagenomic assemblies for short antimicrobial peptide (AMP) open-reading frames. Applying AMPLIT to seven well-curated $\approx 1,000$ - to 2,000-year-old human coprolite metagenomes, the authors shortlisted 160 AMP candidates. Forty peptides were synthesized and 36 (90%) inhibited the growth of four pathogens in vitro; roughly two-thirds of the hits originated from *Segatella copri*, a commensal strongly enriched in ancient but not modern guts. Seven *S. copri* peptides with broad-spectrum activity showed $\leq 10\%$ hemolysis and negligible Caco-2 toxicity, and five leads (i.e., AMP8, AMP13, AMP30, AMP31, AMP38) accelerated wound closure and reduced bacterial burden in a topical murine infection model at 100 μM , achieving efficacy comparable to vancomycin against *Staphylococcus aureus*. The interesting study adds to emerging literature in this field and positions ancient microbiomes as an untapped reservoir of AMPs.

Specific comments:

- Activity was screened at 100 μM and the same high dose was used in vivo; This is a very high dose for peptides, limiting their therapeutic potential.
- MICs should be shown quantitatively (through CFU/mL measurements).
- All in vivo data should be quantified (and shown in the main figures), and most (if not all) qualitative results should be moved to the SI.
- This work contributes to a growing body of studies mining ancient samples (e.g., doi: 10.1038/s41551-024-01201-x) and microbiomes (e.g., doi: 10.1016/j.cell.2024.07.027, describing the discovery of prevotellin-2; doi: 10.1016/j.cell.2024.05.013) for novel peptide antibiotics and needs to be described in the context of the existing literature.
- In vitro testing covers only four laboratory strains, and in vivo efficacy was assessed against *S. aureus* and *P. aeruginosa* wound infections. Though not strictly required, testing for activity against multidrug-resistant clinical isolates would add to the study.
- Please move SEM images showing membrane disruption to the SI.
- AMPLIT is only compared to the authors' previous method (AMPIdentifier). This comparison should be extended to other methods (e.g., APEX, AMPSphere, etc.).
- The authors should better highlight the novelty of AMPLIT over AMPIdentifier.
- Candidates were limited to 13–26 amino acids and canonical residues; no effort was made to improve stability (e.g., D-amino acids, cyclization) or to test synergistic combinations suggested by the authors. This needs to be mentioned as a limitation.
- What was the rationale for limiting sequences to 13–26 amino acid residues?
- Several comparisons lack details of statistical tests or corrections for multiple hypotheses, and raw activity data (IC₅₀/MIC tables, hemolysis datasets) have not been deposited in a public repository.
- The manuscript should be thoroughly revised for grammar issues.
- I suggest changing the title to "Ancient gut microbiomes harbor unexplored antimicrobial peptides".

(Remarks on code availability)

Version 1:

Reviewer comments:

Reviewer #1

(Remarks to the Author)

The revised manuscript has improved in clarity, and the authors have added relevant methodological details and study limitations. However, my primary concern regarding the ORF mining strategy remains unaddressed.

1. The existing biological mechanisms of encrypted peptides generation (post-translational cleavage; genomic mutations) cited in the authors' references are inconsistent with their computational approach. The custom script (ORF_hunter.py) defines ORFs solely through promoter and terminator predictions and applies a strict length cutoff (33-150 nt), which is inherently biased. Furthermore, the possibility that the AMPs reported in the cited literature occur naturally is reasonable. Still, the study in the manuscript appears to have mined short sequences that lack transcriptional and translational potential but exhibit moderate antibacterial activity, which may not reflect biological relevance. If these peptides do not correspond to naturally occurring or biologically active entities, their antimicrobial potency may not genuinely support the significance of this study.
2. The authors' own data, showing most contigs are longer than 1 kb, provides a direct opportunity to validate the fragmented origin of these ORFs. Specifically for the AMPs listed in Figure 2a (e.g., AMP12, AMP29, AMP30), did the authors attempt to annotate the larger protein sequences encoded on these same assembled contigs?
3. The study's central contribution needs to be clarified. The authors should decide whether the primary focus is on demonstrating the mining capabilities of the AMPLIT tool itself, or on revealing the AMP potential of ancient gut microbiomes and *S. copri*. This focus should be clearly reflected in targeted modifications to the abstract and main text better to highlight the study's core novelty and findings.

(Remarks on code availability)

Reviewer #2

(Remarks to the Author)

While I thank the authors for their efforts, I feel that some points were not adequately addressed in the rebuttal. Perhaps this is due to my lack of clarity, so I will attempt to clarify my concerns and be more specific.

The authors answer to my concern #2 about using efficiency as a justification seems to misdirect. They claim that their tool is better because it uses more features, but that is not about efficiency. The sentence on L100 still claims "AMPLIT identified putative AMPs within one hour, an efficiency infeasible for previous large pipelines on standard hardware." This is simply not true and the references given do not support it.

With respect to whether their ORF detection is correct (my previous concern #5), I again feel that the authors are answering a different point than the one I was making. Indeed, encrypted peptides can sometimes be found in larger proteins (and be active). Often active peptides can be "trimmed" down to an active core. It could have been interesting to run 6-frame translation on the input, but this is not what their code is doing. To expand on the example given previously, the following sequence codes for AMP10.387_945:

>GMSC10.SMORF.002_433_233_034

ATGTCCTCGCTGGCGATGAGTTCCGCCCTCTCAAGAAGGAAGGCCTGCTGACCCCGACCCAGAAATGAAGGAGCGCAAGAAGTACGGCCTTGAAGGACGCCCGTCCGCGACCCAGTCTCCA

Their code extracts two ORFs from this sequence:

```
ATGAGTTCCGCCCCCTCTCAAGAAGGAAGCCTGCTGA
and
ATGAAGGAGCGCAAGAAGTACGGCTTGAAAGCAGCCCGTCGCGCACCCCAGTTCTCCAAGAGATAA
```

The second one appears to be the shorter fragment that was discussed earlier, that matches the AMP10.387_945 sequence from AMPSphere. As a side-note: this behaviour (outputting embedded ORFs) explains the lower molecular weights observed in AMPLiT predictions compared to other databases, as many peptides will be truncated due to the presence of out-of-frame start codons (cf. the statements on L104+).

However, consider the variation below with a single base change:

```
>GMS10.SMORF.002_433_233_034_variant
ATGCTCCTCGCTGGCGATGAGTTCCGCCCCCTCTCAAGAAGGAAGCCTGCTGACCCGCGACCCAGAAATGAAGGAGCGCAAGAAGTACGGCTTGAAAGCAGCCCGTCGCGCACCCCAGTTCTCCA
```

This changes the penultimate base from A to G, which should be a silent mutation as TAA and TGA are both STOP in the standard table. This sequence codes for the exact same peptide, but now their code fails to extract the second one because the string ATG in the last four bases triggers the start of a new ORF even though it is out of frame!

The 3rd point in their answer is very interesting, but a complete speculation from their data.

Overall, I still feel that the authors have not adequately addressed my concerns. The authors may wish to simply acknowledge that their code inadvertently misses some encrypted peptides due to the way ORFs are extracted, and that this could be improved in future versions.

(Remarks on code availability)
See the comment about ORF_finder.py

Reviewer #3

(Remarks to the Author)
The authors have mostly addressed my prior comments.

(Remarks on code availability)

Version 2:

Reviewer comments:

Reviewer #1

(Remarks to the Author)
Although I appreciate the authors' efforts, I believe that my comments 1-3 were not sufficiently addressed in the rebuttal. For instance, regarding comment 3, the absence of a comparison of MIC values with Vancomycin and the weak activity of the discovered AMPs do not adequately support the claim of the biological significance of using ancient metagenomic samples for novel AMP discovery.

(Remarks on code availability)

Reviewer #2

(Remarks to the Author)
The authors have addressed my previous concerns.

I think there could be more analysis on the point related to the evolution of the genes partially matching other previously reported peptides, but this does not affect the text (it is only mentioned in the rebuttal document) and I recognise it is outside the scope of the current manuscript.

(Remarks on code availability)

Open Access This Peer Review File is licensed under a Creative Commons Attribution 4.0 International License, which permits use, sharing, adaptation, distribution and reproduction in any medium or format, as long as you give appropriate credit to the original author(s) and the source, provide a link to the Creative Commons license, and indicate if changes were made. In cases where reviewers are anonymous, credit should be given to 'Anonymous Referee' and the source.

POINT-BY-POINT REPLY TO EDITORS AND REVIEWERS

Dear Reviewers,

Thank you very much for the valuable comments to our manuscript. We have provided a point-by-point response to the reviewers' comments, together with a tracked and clean version of the revised manuscript. All changes are highlighted.

We appreciate the opportunity to resubmit and hope you find this version acceptable for publication.

Yours sincerely

Qi Su on behalf of co-authors

Reviewer #1:

Remarks to the Author:

1. The manuscript presents a compelling study that utilizes portable hardware-compatible AI (AMPLiT) to mine novel antimicrobial peptides (AMPs) from ancient human coprolite metagenomes, predominantly derived from *Segatella copri*. The identification and subsequent validation of active AMPs against both Gram-positive and Gram-negative pathogens through in vitro and in vivo assays is commendable. The evolutionary rationale for exploring ancient microbiomes as a strategy to overcome modern resistance mechanisms is particularly insightful.

However, it is essential to clarify that the identified AMPs are not traditional secreted bacterial peptides or host-derived AMPs, but rather fragments of various functional bacterial proteins. The rationale behind this approach requires further explanation. Additionally, the study did not include antimicrobial-resistant (AMR) strains in its analyses; consequently, the discovered AMPs did not demonstrate potent activity against AMR pathogens. This limitation should be explicitly acknowledged.

To enhance the manuscript's rigor and impact, the authors should benchmark AMPLiT against existing AMP discovery methods to contextualize its performance and advantages. While the manuscript is generally well-structured, several key points—such as the methodology, rationale, and experimental scope—would benefit from clarification and expansion. Furthermore, the figure legends are currently insufficiently detailed and difficult to read; improving their clarity and comprehensiveness would greatly aid reader understanding.

Response: We thank the reviewer for the positive comments. We have revised the manuscript

as suggested.

1. Major Comments

1) Many AMPs were found to be fragments of functional proteins.

Response: We agree that this is an important point to clarify. Indeed, many of the putative AMPs identified in our study are fragments derived from functional bacterial proteins. This phenomenon is not uncommon in AMP discovery. The underlying rationale is that certain regions of bacterial proteins can be ancestors of AMP genes and may exhibit intrinsic antimicrobial activity. This has been previously observed and reported in other studies of novel AMP discovery ^[1], in which modern and extinct human proteins were found to harbor antimicrobial subsequences. These cryptic antimicrobial regions may serve as an evolutionary reservoir for host defense or inter-microbial competition ^{[2]-[3]}. Specifically, the reference [2] involves a whole section named “*Mutations in larger genes generate c_AMPs as independent genomic entities*” to support the evolutionary origin of AMPs from larger functional proteins and authors stated that “*we hypothesized that some of these have originated from larger proteins by fragmentation at the genomic level*”. These previous studies indicated the underlying rationale of the AMPs’ origin.

In the context of our study, which focuses on ancient microbiome data, it is plausible that such fragments could have represented a form of microbial arsenal or regulatory molecule in complex ancestral communities. Their identification through AMPLiT suggests that our model recognizes such sequences with potential antimicrobial properties.

[1] Molecular de-extinction of ancient antimicrobial peptides enabled by machine learning, *Cell Host & Microbe*, 2023, 31, 8, 1260-1274.e6.

[2] Discovery of antimicrobial peptides in the global microbiome with machine learning. *Cell*, 2024, 187, 3761-3778.e3716.

[3] Antimicrobial host defence peptides: functions and clinical potential. *Nature Reviews Drug Discovery*, 2020, 19, 311-332.

2) AMPLiT against existing AMP discovery methods to further contextualize advantages

Response: Thank you again for your insightful suggestion, and we have supplemented this part to further contextualize advantages of AMPLiT. By using the same training and testing dataset from the recently published study ^[2], the comparison between AMPLiT and a series of the recently proposed state-of-the-art models ^[2] has been included in **Table 1**. AMPLiT with low computational burdens showed robust performances in multiple indicators comparable to PGAT-ABPp and other methods (**Table 1**). PGAT-ABPp highly relies on the accuracy of the predicted peptide structures, potentially biased by inaccurate structural prediction risks and high computational burdens.

While PGAT-ABPp push the boundaries of performances, it requires significant GPU resources and are not validated by *in vitro* and *in vivo* experiments. Based on these results, AMPLiT demonstrates a compelling combination of reasonable computational efficiency and high predictive performances.

Table 1. Comparison of the AMPLiT and PGAT-ABPp using the same training/testing data from ^[1]

Method	Specificity (%)	Precision (%)	Accuracy (%)	AUC	References
AMPLiT	97.26 ± 0.67	95.38 ± 1.03	96.69 ± 0.46	0.9910 ± 0.0017	In this study
AMPIdentifier	96.46 ± 0.86	94.03 ± 0.99	95.40 ± 0.77	0.9872 ± 0.0014	[1]
PGAT-ABPp	96.72 ± 0.38	95.31 ± 0.51	96.49 ± 0.21	0.9936 ± 0.0007	[2]
Onehot-GAT	96.49 ± 0.21	87.33 ± 3.90	92.15 ± 0.90	0.9776 ± 0.0039	[2]
Word2vec-GAT	82.85 ± 5.62	79.83 ± 5.17	88.25 ± 3.26	0.9623 ± 0.0142	[2]
ProtT5-CNN	94.37 ± 0.28	92.30 ± 0.36	95.61 ± 0.16	0.9924 ± 0.0005	[2]

[1] Screening and identification of antimicrobial peptides from the gut microbiome of cockroach *Blattella germanica*. *Microbiome*, 2, 272.

[2] PGAT-ABPp: harnessing protein language models and graph attention networks for antibacterial peptide identification with remarkable accuracy. *Bioinformatics*, 40, 8, btae497.

3) Explanations to other mentioned concerns

Response: According to the valuable suggestions from reviewer, we have thoroughly revised the manuscript to provide more clarity and details regarding the methodology, rationale, and experimental scope. We sincerely hope these efforts could enhance the transparencies and clarities of the manuscript. Assessing the activity of our lead AMPs against multidrug-resistant (AMR) clinical isolates is an important aspect of our future researches. We fully acknowledge the critical importance of experimentally evaluating the efficacy of these peptides against contemporary AMR pathogens, and we have explicitly highlighted it as a key and necessary goal for our future research (Line 185-187). Accordingly, our on-going project plans to expand the screening of identified peptides to a broader panel of clinically relevant multidrug-resistant strains.

The description of the AMPLiT framework, including its architecture, training process, and hyper-parameters optimization steps, to improve reproducibility, has been further supplemented and explained (Methods section). The evolutionary and ecological premises for mining AMPs from ancient microbiomes have been more thoroughly articulated in the manuscript (Line 179-194). All figure legends have been revised to be more comprehensive, with improved explanations of experimental designs, statistical analyses, and panel interpretations to enhance readability and contextual understanding (Lin 403-464).

Minor comments:

1. While the in vitro cytotoxicity results are presented, the potential for these evolutionarily distant AMPs to trigger heightened immunogenicity in a modern mammalian host (e.g., humans) warrants serious consideration. If feasible, preliminary cytokine profiling using human immune cells in vitro would significantly strengthen the safety profile assessment. At minimum, this must be highlighted as a key consideration for future development.

Response: Thank you for the important suggestion and we sincerely agreed with the importance of emphasizing “cytokine profiling using human immune cells” as a key consideration for future development. According to your requests, we have revised our manuscript and include the discussion of necessities of conducting the preliminary cytokine profiling using human immune cells in future researches and investigations (Line 187-189).

2. The paper focuses on pathogen suppression but ignores a key aspect: potential collateral damage to the host's beneficial microbiota. The antimicrobial peptides (AMPs) discovered by the authors are broad-spectrum, and their impact on modern commensal bacteria community is unclear.

Response: We highly appreciate this valuable suggestion and we agreed with the mentioned consideration of key aspect in AMPs' influences to host's beneficial microbiota. Although the AMPs we discovered from *S. copri* exhibited broad-spectrum antimicrobial activity, these AMP ORFs have also been conservatively reserved in genomes of *S. copri* assembled from modern populations living in rural areas. These phenomena indirectly highlight the potential low safety concerns to modern commensal community, as they still exist in modern humans despite their low prevalence. Lastly, we sincerely acknowledged the limitations of not assessing the potential influences of these AMPs to commensal community in this study, and we have clarified it as a key goal for future research in the manuscript (Line 187-189).

3. Several references appear to be incorrectly cited, for example in Line 104, reference 19 is not relevant to toxicity prediction. In Line 123, references 7 and 17 do not seem to support the author's point of view. Please verify and replace with an appropriate citation.

Response: Thank you so much for the kind reminds, and please allow us to clarify.

In details, the reference 19 proposed the tool of “ToxinPred2”, which enables the prediction of peptide cytotoxicity, and we have utilized it in this study for excluding those peptides with potential cytotoxicity.

For references of 7 and 17, these 2 literatures indirectly support the view of “AMPs origin”. For example, the reference 17 include a whole section named “*Mutations in larger genes*”.

generate c_AMPs as independent genomic entities” to support the evolutionary origin of AMPs from larger functional proteins. And authors of reference 17 stated that *“we hypothesized that some of these have originated from larger proteins by fragmentation at the genomic level”*. In reference 7, modern and extinct human proteins were found to harbor antimicrobial subsequences. These cryptic antimicrobial regions may serve as an evolutionary reservoir for host defense or inter-microbial competition.

To further alleviate concerns from the reviewer, we have supplemented 2 additional references to support our viewpoints. In literature [1]-[2], they mention that the AMP gene may originate from a fragment of a larger gene. Combined with the original references of 7 and 17, these two references have also been supplemented in the Line 126 of the manuscript. And we hope these efforts can alleviate the concerns from reviewer.

[1] Antimicrobial host defence peptides: functions and clinical potential. *Nat Rev Drug Discov*, 19, 311–332.

[2] Cryptic Antimicrobial Peptides: Identification Methods and Current Knowledge of their Immunomodulatory Properties. *Curr Pharm Des*, 24, 1054-1066.

4. More details could be added on the development of AMPLiT, such as to specify the transformer architecture used.

Response: Thank you so much for the important suggestion, and we have supplemented more technical details of the AMPLiT architecture in the manuscript according to your suggestion. For improving the transparency of the manuscript, we have also added more technical details in the “Method” section (Line 195-369).

5. Did the authors predict AMPs from the metagenome assembled contigs or MAGs? And in Fig 1h and Line 112, how did the authors assign hosts to these AMPs? Please add this detail in the methods.

Response: Thank you so much for this valuable question, and we have revised this detail in the Method section (Line 252-264). And please allow us to answer your questions in details.

1) Are these AMPs from the metagenome assembled contigs or MAGs?

These AMPs were identified within the assembled metagenomic contigs, which were assigned to ancient prevalent gut microbiota *S. copri*.

2) how did we assign hosts to these AMPs?

According to the figure (left panel) shown below, these contigs are assigned to host *S. copri* by explicitly aligning to those assembled contigs to the reference genomes deposited in NCBI. Due to erosion of the microbial DNA in ancient samples (left panel in the figure below), their contigs were not able to fully recover the complete genome of the *S. copri* and can only annotate at the species level (*S. copri*). Fortunately, the ancient *S. copri*-derived contigs containing these AMP's ORFs possess sufficient sequencing reads coverage (right panel below), indicating high assembly quality and good preservation of these contigs despite the long history of samples (1000-200 years ago).

Furthermore, the contig (containing those identified AMP ORFs) alignment also shared astonishing high conservations to those *S. copri* genome isolated from modern rural populations (~94-99% similarity, Figure below) in NCBI. These observations clearly traced the potential molecular origins of these AMPs from *S. copri*.

6. Line 241, the instability index is not shown in the figure.

Response: Thank you so much for the careful checking, and we have deleted this typing error in the current version of the manuscript accordingly (Line 411-413).

7. Figure 2d, why does AMP31 or other AMPs have an inhibition rate of 100% at a concentration of 25 mM, but the inhibition rate decreases at higher concentrations?

Response: Thank you so much for this valuable question, and we also noticed these interesting phenomena. As a natural property observed in many peptides (including AMPs [1]), we speculate that the AMP molecules may undergo self-aggregation or form oligomers under high dosage concentrations. The self-aggregation may cause these peptides not able to show the same antimicrobial killing efficacies as they were under lower dosage, thereby potentially resulting in a decreased result.

[1] Computational Methods and Tools in Antimicrobial Peptide Research. *Journal of Chemical Information and Modeling*, 61, 3172-3196.

8. The *in vivo* results are presented qualitatively, such as representative images. The lack of quantification severely limits interpretation and statistical validation.

Response: Thank you so much for this valuable question. We have conducted the *in vivo* quantitative analysis before, and here we supplemented the results below. These are results of the decreased bacteria load in infected tissues.

As previously stated in “Methods” section, we immersed the wound granulation tissues by PBS solutions, and split them into two sections for H&E/Masson staining and CFU load observations. In addition to the CFU load observations (Extended Figure 4b), the supernatant solutions immersed with the infected wound granulation tissues were further centrifugated, and we quantified the OD₆₀₀ values of the supernatant solutions to further assess and calculate the decreased bacteria load (supplemented in Figure 2 and Methods). The quantitative results supported the potential therapeutic efficacy of these identified AMPs.

In addition, the quantitative results of measuring wound closure size at Day 12 across different experimental groups are also supplemented here (Figure above, supplemented in Figure 2 and Methods). Taken together, these results also suggested potential therapeutic efficacy of AMP13, AMP30, and AMP38 *in vivo*.

Hope our efforts aforementioned can alleviate your concerns, and thank you so much for the valuable suggestions.

Reviewer #2 (Remarks to the Author):

This manuscript from Chen, Yuan, et al. presents several AMPs extracted from ancient gut microbiome data. They validate several candidates *in vitro*, with a high hit rate (albeit by using a high threshold of 100 μ M to define positive, see #3 below) and test the top candidate *in vivo* (but see #6 below).

Overall, I find that the work could be very interesting but several details are not clear enough to make a full assessment and some claims should be toned down.

Response: We thank the reviewer for the positive comments. We have revised the manuscript as suggested.

MAJOR COMMENTS

1. Could the authors provide the codes in a more organized fashion? The authors mention that "All code to reproduce data analyses in this Brief Communication is provided in the Supplementary Information", but there is still a bit of a need to follow links to previous papers and repositories and nowhere is everything put together. At least a single document with instructions would be very helpful. In particular, I failed to find the code that produces the alignments shown in Fig 2A.

Response: Thank you so much for this valuable suggestion. According to your suggestions, we have revised our manuscript accordingly to include more technical and coding details in "Methods" section (Line 195-357) for avoiding the need to follow links to previous papers and repositories. In addition, we have also revised the coding repository at <https://github.com/ChenSizhe13893461199/AMPLiT> to put everything together with expanded instructions. For information and the alignments shown in Fig 2A, we are sorry for the inconvenience this has caused. Here are no particular codes used for this analysis, as it was

mainly finished by using online tools from public websites. For your convenience, here we provided the full ORF sequences of each AMPs that were analyzed in Figure 2 at the aforementioned coding repository (AMP_S.copri_ORF.xlsx). By using these data, the figure and data can be easily reproduced via tools provided by public websites (e.g. NCBI BLAST).

2. The argument of computational efficiency is not very convincing. In all these pipelines, the bottleneck is—by far—the assembly and not the AMP prediction step. For example, I can run the MACREL tool (which the authors indirectly cite as support for their claim as it is the tool behind [17]) on thousands of peptides per second (single CPU, using a laptop that was modern in 2021).

Response: We sincerely thank the reviewer for this insightful comment. We fully agree that metagenomic assembly is indeed a computationally intensive step that often constitutes the bottleneck in traditional AMP discovery pipelines, especially for ancient or highly fragmented samples.

The primary motivation behind developing AMPLiT was not to accelerate the assembly step, but rather to provide a lightweight, hardware-accessible, and efficient tool for rapid model training, and accurate downstream AMP screening. While tools like MACREL seems to be efficient for peptide-level screening, MACREL only utilized 22 physiochemical features for model training and AMPs screening. As a comparison, AMPLiT enables rapid training within 3200 ± 53 seconds and incorporates more than 3,547 physiochemical features for each input peptide sequences (1,000 features for one-hot code compositional information, 1,547 features for physiochemical properties, 1,000 features for Word2Vec embedding features), which is 161 times greater in scale than the previous tool MACREL. This potentially enhanced the screening performances of AMPLiT.

Table 1. Comparison of the AMPLiT and PGAT-ABPp using the same training/testing data from ^[1]

Method	Specificity (%)	Precision (%)	Accuracy (%)	AUC	References
AMPLiT	97.26 ± 0.67	95.38 ± 1.03	96.69 ± 0.46	0.9910 ± 0.0017	In this study
AMPIdentifier	96.46 ± 0.86	94.03 ± 0.99	95.40 ± 0.77	0.9872 ± 0.0014	[1]
PGAT-ABPp	96.72 ± 0.38	95.31 ± 0.51	96.49 ± 0.21	0.9936 ± 0.0007	[2]
Onhot-GAT	96.49 ± 0.21	87.33 ± 3.90	92.15 ± 0.90	0.9776 ± 0.0039	[2]
Word2vec-GAT	82.85 ± 5.62	79.83 ± 5.17	88.25 ± 3.26	0.9623 ± 0.0142	[2]
ProtT5-CNN	94.37 ± 0.28	92.30 ± 0.36	95.61 ± 0.16	0.9924 ± 0.0005	[2]

In addition, we have also supplemented the comparison results with other recently proposed state-of-the-art models ^[2] to further contextualize advantages of AMPLiT (**Table 1** above). By using the same training and testing dataset from the recently published study ^[2], the comparison between AMPLiT and a series of those recently proposed state-of-the-art models ^[2] has been included in

Table 1. AMPLiT with low computational burdens showed robust performances in multiple indicators comparable to PGAT-ABPp and other methods (**Table 1**). PGAT-ABPp highly relies on the accuracy of the predicted peptide structures, potentially biased by inaccurate structural prediction risks and high computational burdens. While PGAT-ABPp push the boundaries of performances, it requires significant GPU resources and are not validated by *in vitro* and *in vivo* experiments. Based on these results, AMPLiT demonstrates a compelling combination of reasonable computational efficiency and high predictive performances.

[1] Screening and identification of antimicrobial peptides from the gut microbiome of cockroach *Blattella germanica*. *Microbiome*, 2, 272.

[2] PGAT-ABPp: harnessing protein language models and graph attention networks for antibacterial peptide identification with remarkable accuracy. *Bioinformatics*, 40, 8, btae497.

3. A threshold of 100 μM is high for AMP determination. For the ones that the authors follow up on, they do use lower thresholds, but it would be good to either see the data for all AMPs or at least mention the high threshold already in the abstract (when the authors mention the 90% hit rate).

Response: Thank you so much for this valuable suggestion, and we have revised the abstract to include the threshold of 100 μM dosage in the abstract accordingly.

4. The lower molecular weight of the peptides here compared to the previously reported ones could be due to differences in origin species. Several peptides in the databases are from vertebrate origin, which may influence the comparison. I am also do not find the inference that this may indicate potential membrane-penetration activity (L100) very convincing. Most AMPs are anyway membrane-disrupting, so is the argument here that the ancient ones are even more likely to be active in this way? In any case, the relationship with low microbial weight is not a strong one.

Response: Thank you for this insightful comment. We agree that molecular weight alone is not a strong predictor of function and that the comparison to the reference database could be confounded by origin. Therefore, we have revised the manuscript to remove any speculative link between molecular weight and membrane-penetration activity. The sentence now simply reports the lower molecular weight as an objective characteristic of the filtered candidate set, without implying a functional consequence.

5. Also, if I understand the functionality of ORF_finder.py correctly (I'm reading the version here https://github.com/ChenSizhe13893461199/Fast-AMPs-Discovery-Projects/blob/2e1147625a96a6133875c309e5418882b9663d44/ORF_hunter.py which may not be

the right version, but see #1), the authors actually consider the shortest possible ORFs even inside another longer ORF! It is such an odd choice that I would tend to consider it a bug in the code. Also, if there is a out-of-frame START, it will reset the scanning to consider a different frame. This also affects any claims of novelty. For example, taking just their first sequence (KERKKYGLKAARRAPQFSKR), this matches a longer peptide in AMPSphere: https://ampsphere.big-data-biology.org/amp?accession=AMP10.387_945 but given the way that ORFs are called here, only the shorter fragment is considered.

Response: We thank the reviewer for this insightful comment and for closely examining our ORF detection methodology. The reviewer has accurately identified a key feature of our tool, ORF_finder.py: it is indeed designed to consider short ORFs nested within longer ones and to initiate new ORF scanning upon encountering a start codon in any reading frame.

1) We wish to clarify that this is not a bug but a deliberate design choice, tailored specifically for the *de novo* discovery of AMPs from ancient metagenomes. This approach is grounded in the established evolutionary biology of AMPs. A growing body of research indicates that novel AMP genes frequently originate from the fragmentation and exaptation of larger housekeeping genes ^{[1]-[5]}. For instance, it has been shown that human proteins harbor cryptic antimicrobial regions ^[1], potentially contributing to novel AMP's origin. And a recent study dedicated a section "*Mutations in larger genes generate c-AMPs as independent genomic entities*" and they stated that "*we hypothesized that some of these have originated from larger proteins by fragmentation at the genomic level*" ^[2].

2) This strategy is also crucial for analyzing potentially fragmented ancient DNA entities (~1000-2000 years old), where the recovery of full-length genes is exceedingly rare compared to modern samples. By design, our method aims at maximizing the probability of detecting all possible translation initiation events, thereby identifying short, complete ORFs that have a higher chance of encoding functional peptides.

3) Lastly, the AMP1 identified by AMPLiT is a subset of the longer AMP10.387_945 (40 aa) found in AMPSphere. However, our peptide is sourced from an ancient human commensal while the AMPSphere's entry is derived from *Ruminococcaceae* genus of the modern cattle gut. This phylogenetic and temporal distinction is significant. And we experimentally validated the antimicrobial activity of AMP1 *in vitro*. This confirms that this specific fragment is itself a functional antimicrobial unit, irrespective of the activity of the longer peptide it is embedded within. AMPLiT potentially provides insights in identifying active fragments that may not be annotated in modern databases focused on full-length proteins. It reveals that this specific bioactive fragment was encoded and potentially functional over a thousand years ago in a

different host environment, providing a new evolutionary perspective that is not captured by modern databases alone.

We hope these explanations could alleviate the reviewer's concerns, and thank you again for your valuable comment.

[1] Molecular de-extinction of ancient antimicrobial peptides enabled by machine learning, *Cell Host & Microbe*, 2023, 31, 8, 1260-1274.e6.

[2] Discovery of antimicrobial peptides in the global microbiome with machine learning. *Cell*, 2024, 187, 3761-3778.e3716.

[3] Antimicrobial host defence peptides: functions and clinical potential. *Nature Reviews Drug Discovery*, 2020, 19, 311-332.

[4] Antimicrobial host defence peptides: functions and clinical potential. *Nat Rev Drug Discov*, 19, 311–332.

[5] Cryptic Antimicrobial Peptides: Identification Methods and Current Knowledge of their Immunomodulatory Properties. *Curr Pharm Des*, 24, 1054-1066.

6. The claims related to the distribution of the *S. copri* AMPs in different strains (L130) are potentially very interesting, but it is not clear where the strain genomes are coming from. If the authors are using modern genomes, then this partly disproves the value of using ancient data (although it is very interesting from an evolutionary perspective to see that these genes have been conserved). In any case, the authors should provide more details on the genomes used.

Response: Thank you for this valuable question and comment, and we have revised to supplement more details in the Method section (Line 252-264). These AMPs were identified within the assembled metagenomic contigs, which were assigned to anciently prevalent gut microbiota *S. copri*. According to the figure (left panel) shown below, these contigs are assigned to host *S. copri* by explicitly aligning to those assembled contigs to the reference genomes deposited in NCBI. Due to erosion of the microbial DNA in ancient samples (left panel in the figure below), their contigs were not able to fully recover the complete genome of the *S. copri* and can only

annotate at the species level (*S. copri*). Fortunately, the *S. copri*-derived contigs containing these AMP's ORFs possess sufficient sequencing reads coverage (right panel), indicating high assembly quality and good preservation of these contigs despite the long history (1000-200 years ago).

Furthermore, the contigs (containing those identified AMP ORFs) alignment also shared astonishing high conservations to those *S. copri* genome isolated from modern rural populations (~94-99% similarity, Figure below) in NCBI. These observations clearly traced the potential origins of these AMPs from *S. copri*.

7. The *in vivo* results are potentially very important, but there is a lack of quantification. The data is shown as an extended figure and it is, frankly, hard to be perform all the necessary comparisons by eye (and only a single replicate of each condition is shown). The authors should present the CFU counts as well and quantify the wound sizes in a systematic way.

Response: Thank you so much for this valuable question. Though the CFU load observations

can effectively prove the efficacy of our AMPs, the high density of the bacteria colonies in negative control groups makes it currently hard to obtain quantitative counting comparisons. And here we supplemented additional quantitative results of assessing the decreased bacteria load in infected tissues (Figure below).

As previously stated in “Methods” section, in our previous experiments, we immersed the wound granulation tissues by 2 mL PBS solutions, and split them into two sections for H&E/Masson staining and CFU load observations. In addition to the CFU load observations (Extended Figure 4b), the supernatant solutions immersed with the infected wound granulation tissues were further centrifuged, and we quantified the OD₆₀₀ values of the supernatant solutions to further assess and calculate the decreased bacteria load (supplemented in Figure 2 and Methods). The quantitative results supported the potential therapeutic efficacy of these identified AMPs.

In addition, the quantitative results of measuring wound closure size at Day 12 across different experimental groups are also supplemented here (Figure above, supplemented in Figure 2 and Methods). Taken together, these results also suggested potential therapeutic efficacy of AMP13, AMP30, and AMP38 *in vivo*.

Hope our efforts aforementioned can alleviate your concerns, and thank you so much for the valuable suggestions.

MINOR COMMENTS

8. Finding *S. copri* as a high producer of AMPs is consistent with previous literature, see DOI 10.1016/j.cell.2024.05.013 or DOI 10.1016/j.cell.2024.07.027 (which use the name *Prevotella copri*).

Response: Thank you so much for this valuable remind and comment, and we have supplemented these references and clearly claimed their discoveries in our manuscript (Line 66-73). These literatures identified AMPs from *Prevotella* genus (e.g. *Prevotella jejuni*, and *Prevotella melaninogenica*, implying that *S. copri* might also be a high producer of AMPs).

9. Figs 1c-e: I would ask that the authors use box and dot plots instead of the dynamite ones which do not show the distribution of data well.

Response: Thank you so much for this valuable question, and we have revised the manuscript according to your requests for showing the data distribution. Due to the huge amounts of the data size, we choose to draw the data by box and violin plots, which might be clearer in representing data distribution pattern (Figure 1c-1e).

10. L113: what is the criterion behind calling some AMPs "highly potent"?

Response: Thank you so much for this valuable question and remind. Here we have revised the manuscript to include the criterion behind calling some AMPs "potent" (Line 115-118). Specifically, we define those AMPs with at least 50% inhibitory rate to at least one tested bacterial strain to be potent candidates, and we have included this explanation in the current manuscript.

Reviewer #2 (Remarks on code availability):

See the remark on the main review: the code is only partially available or, at least it was not clear to me how to access all of it. Some important questions are not clear from the Methods/Figure legends and without the code, it was impossible to fully assess the paper.

Response: Thank you so much for this valuable question, and here we have revised the manuscript "Methods" section (Line 195-357) and the coding repository at the GitHub (<https://github.com/ChenSizhe13893461199/Fast-AMPs-Discovery-Projects/>) to include more technical details of AMPLiT and put everything together without the need to follow links to previous papers. The revised content includes more details of utilization guidance, framework mechanisms, model training and assessment methods, and full access to codes. We sincerely hope our efforts can alleviate the concerns from reviewer.

Reviewer #3 (Remarks to the Author):

This paper by Chen et al. introduces AMPLiT, a deep-learning pipeline that can be run on consumer-grade CPUs to mine metagenomic assemblies for short antimicrobial peptide (AMP) open-reading frames. Applying AMPLiT to seven well-curated \approx 1,000- to 2,000-year-old human coprolite metagenomes, the authors shortlisted 160 AMP candidates. Forty peptides were synthesized and 36 (90%) inhibited the growth of four pathogens in vitro; roughly two-thirds of the hits originated from *Segatella copri*, a commensal strongly enriched in ancient but not modern guts. Seven *S. copri* peptides with broad-spectrum activity showed $\leq 10\%$ hemolysis and negligible Caco-2 toxicity, and five leads (i.e., AMP8, AMP13, AMP30, AMP31, AMP38) accelerated wound closure and reduced bacterial burden in a topical murine infection model at 100 μ M, achieving efficacy comparable to

vancomycin against *Staphylococcus aureus*. The interesting study adds to emerging literature in this field and positions ancient microbiomes as an untapped reservoir of AMPs.

Specific comments:

1. Activity was screened at 100 μM and the same high dose was used *in vivo*; This is a very high dose for peptides, limiting their therapeutic potential.

Response: We thank the reviewer for raising this important point regarding the peptide concentration used in our studies, and please allow us to explain in details.

For *in vitro* screening, the use of a single, relatively high concentration (e.g. 100 μM) for the initial primary *in vitro* screening is a frequently-utilized strategy ^{[1]-[2]}, allowing for the efficient identification of peptides from a large pool of candidates (40 synthesized peptides in our case). As the reviewer rightly implies, we did not stop at the 100 μM screen and we performed comprehensive dose-response assays for the most promising leads (Fig. 2d), providing the critical data on potency and therapeutic potential.

For *in vivo* experiment, we conducted our *in vivo* wound healing experiment based on the following considerations:

1) The AMPs identified in our study were locally administrated and this manner allows for the use of higher concentrations ^{[1], [3]} that would be unsuitable for systemic delivery, as the local administration minimizes the risk of systemic toxicity (and we have also proved their minimal hemolytic activity and low cytotoxicity in Extended Data Figure 3 a-h).

2) We benchmarked our peptides against standard antibiotics (vancomycin and polymyxin B) and lead peptides performed comparably to these antibiotics under identical conditions highlights their efficacy and potential.

3) Here we attempted to provide a proof-of-concept for AMPs derived from ancient gut microbiota. We fully acknowledge that future work needs to focus on chemical engineering and peptide design to further optimize the therapeutic dosages of these identified AMPs. These efforts will contribute to the required therapeutic dosage and improve the clinical potency.

Thank you again for the valuable comment and suggestion.

[1] Identification of antimicrobial peptides from the human gut microbiome using deep learning. *Nature Biotechnology*, 40, 921-931.

[2] Molecular de-extinction of ancient antimicrobial peptides enabled by machine learning. *Cell Host & Microbe*, 31, 1260-1274.e1266.

[3] A sonosensitive diphenylalanine-based broad-spectrum antimicrobial peptide. *Nature*

2. MICs should be shown quantitatively (through CFU/mL measurements).

Response: Thank you so much for this important comment, and here we further conducted MIC measurement experiments by CFU/mL methods (Figure S1c), with results shown below:

MIC of representative lead peptides measured by CFU plate spreading method (μM)

AMP	E. coli 25404	S. aureus 6538	P. aeruginosa 27853
AMP 8	25	100	-
AMP13	10	-	100
AMP30	25	5	100
AMP31	100	>100	>100
AMP38	25	50	>100

The relevant experimental results have now been thoroughly incorporated into the manuscript (Figure S1c and Table S3). Together, these results suggested that most of the experimental results by CFU methods showed consistency with the previous OD-based assays.

Specifically, AMP8, AMP13, AMP30, AMP31 and AMP38 exhibit bactericidal effects against pathogenic *E. coli* 25404, with results consistent with the previous OD-based assays. For AMPs against *S. aureus* 6538, AMP8, AMP30, and AMP38 also showed bactericidal effects that are consistent with the trends observed in OD-based assays. For AMPs against *P. aeruginosa* 27853, results of AMP13, AMP30 and AMP 31 generally showed consistency with the previous OD-based assays. Interestingly, AMP31 against *S. aureus* 6538 and AMP38 against *P. aeruginosa* 27853 exhibit reduced efficacy on solid media compared to the OD-based liquid culture. This could be due to the potential instability under certain environmental conditions, or it may primarily function as a bacteriostatic agent rather than a bactericidal one in this

particular context. Overall, CFU-derived results are in general alignment with the OD-based inhibitory data in our manuscript.

3. All *in vivo* data should be quantified (and shown in the main figures), and most (if not all) qualitative results should be moved to the SI.

Response: Thank you so much for this valuable question. We have conducted the *in vivo* quantitative analysis before, and here we supplemented the results below. These are results of the decreased bacteria load in infected tissues.

As previously stated in “Methods” section, we immersed the wound granulation tissues by PBS solutions, and split them into two sections for H&E/Masson staining and CFU load observations. In addition to the CFU load observations (Extended Figure 4b), the supernatant solutions immersed with the infected wound granulation tissues were further centrifuged, and we quantified the OD₆₀₀ values of the supernatant solutions to further assess and calculate the decreased bacteria load (supplemented in Figure 2 and Methods). The quantitative results supported the potential therapeutic efficacy of these identified AMPs.

In addition, the quantitative results of measuring wound closure size at Day 12 across different experimental groups are also supplemented here (Figure above, supplemented in Figure 2 and Methods). Taken together, these results also suggested potential therapeutic efficacy of AMP13,

AMP30, and AMP38 *in vivo*.

Lastly, according to your suggestion, we have moved most qualitative results to SI accordingly. Hope our efforts aforementioned can alleviate your concerns, and thank you so much for the valuable suggestions.

4. This work contributes to a growing body of studies mining ancient samples (e.g., doi: 10.1038/s41551-024-01201-x) and microbiomes (e.g., doi: 10.1016/j.cell.2024.07.027, describing the discovery of prevotellin-2; doi: 10.1016/j.cell.2024.05.013) for novel peptide antibiotics and needs to be described in the context of the existing literature.

Response: Thank you so much for this kind suggestion, and we have supplemented these references for describing the context of the existing literature accordingly (Line 66-73).

5. *In vitro* testing covers only four laboratory strains, and *in vivo* efficacy was assessed against *S. aureus* and *P. aeruginosa* wound infections. Though not strictly required, testing for activity against multidrug-resistant clinical isolates would add to the study.

Response: Thank you so much for this valuable suggestion. Assessing the activity of our lead AMPs against multidrug-resistant clinical isolates is a fascinating and important next step for our future research. The primary objective of this study was to determine whether ancient human coprolites, as an underexplored paleogenetic resource, could serve as a viable source for novel and functional antimicrobial peptides. Our goal was to build the entire pipeline from ancient DNA extraction to functional validation, establishing that these 'molecular fossils' can be resurrected and exhibit bioactivity. According to your suggestions, we explicitly emphasized the importance of assessing the efficacy of these peptides against contemporary antimicrobial-resistant (AMR) pathogens and we have clarified this as a key goal for future (Line 184-187). And we will pursue these key aspects in future investigations.

6. Please move SEM images showing membrane disruption to the SI.

Response: Thank you so much for this valuable suggestion, and we have revised to move SEM images showing membrane disruption to the SI accordingly.

7. AMPLiT is only compared to the authors' previous method (AMPIdentifier). This comparison should be extended to other methods (e.g., APEX, AMPSPHERE, etc.).

Response: Thank you again for your insightful suggestion, and we have supplemented this part to

further contextualize advantages of AMPLiT. By using the same training and testing dataset from the recently published study [2], the comparison between AMPLiT and a series of recently proposed state-of-the-art models [2] has been included in **Table 1**. AMPLiT with low computational burdens showed robust performances in multiple indicators comparable to PGAT-ABPp and other methods (**Table 1**). PGAT-ABPp highly relies on the accuracy of the predicted peptide structures, potentially biased by inaccurate structural prediction risks and high computational burdens. While PGAT-ABPp push the boundaries of performances, it requires significant GPU resources and are not validated by *in vitro* and *in vivo* experiments. Based on these results, AMPLiT demonstrates a compelling combination of reasonable computational efficiency and high predictive performances.

Table 1. Comparison of the AMPLiT and PGAT-ABPp using the same training/testing data from [1]

Method	Specificity (%)	Precision (%)	Accuracy (%)	AUC	References
AMPLiT	97.26 ± 0.67	95.38 ± 1.03	96.69 ± 0.46	0.9910 ± 0.0017	In this study
AMPIdentifier	96.46 ± 0.86	94.03 ± 0.99	95.40 ± 0.77	0.9872 ± 0.0014	[1]
PGAT-ABPp	96.72 ± 0.38	95.31 ± 0.51	96.49 ± 0.21	0.9936 ± 0.0007	[2]
Onehot-GAT	96.49 ± 0.21	87.33 ± 3.90	92.15 ± 0.90	0.9776 ± 0.0039	[2]
Word2vec-GAT	82.85 ± 5.62	79.83 ± 5.17	88.25 ± 3.26	0.9623 ± 0.0142	[2]
ProtT5-CNN	94.37 ± 0.28	92.30 ± 0.36	95.61 ± 0.16	0.9924 ± 0.0005	[2]

[1] Screening and identification of antimicrobial peptides from the gut microbiome of cockroach *Blattella germanica*. *Microbiome*, 2, 272.

[2] PGAT-ABPp: harnessing protein language models and graph attention networks for antibacterial peptide identification with remarkable accuracy. *Bioinformatics*, 40, 8, btae497.

8. The authors should better highlight the novelty of AMPLiT over AMPIdentifier.

Response: We thank the reviewer for this critical suggestion, and we have revised the manuscript to highlight the novelty of AMPLiT over AMPIdentifier. We have supplemented our analysis with a comprehensive comparison to other recently proposed state-of-the-art methods, including PGAT-ABPp, Onehot-GAT, Word2vec-GAT, and ProtT5-CNN in the Table above (shown in the previous response). As shown, AMPLiT achieves robust performance, comparable to the current top-performing method (PGAT-ABPp) and superior to several others, while maintaining its unique advantages. AMPIdentifier was a proof-of-concept model that, like PGAT-ABPp and ProtT5-CNN, relied on computationally intensive architectures. PGAT-ABPp highly relies on the accuracy of the predicted peptide structures, potentially biased by inaccurate structural prediction risks and high computational burdens. Furthermore, PGAT-ABPp requires significant GPU resources and are not validated by *in vitro* and *in vivo* experiments. As a comparison, AMPLiT achieved an accessible training time costs on portable equipment, while maintaining its potent performances. We sincerely

hope our efforts aforementioned can alleviate your concerns, and thank you so much for the valuable suggestions.

9. Candidates were limited to 13–26 amino acids and canonical residues; no effort was made to improve stability (e.g., D-amino acids, cyclization) or to test synergistic combinations suggested by the authors. This needs to be mentioned as a limitation.

Response: We sincerely appreciate the reviewer for this critical suggestion, and we have revised the manuscript to clarify the aforementioned limitation accordingly (Line 189-191).

10. What was the rationale for limiting sequences to 13-26 amino acid residues?

Response: We sincerely appreciate the reviewer for asking this insightful question, and we have revised the manuscript to include more details of it (Line 103-107). The decision to limit sequences to certain length scope was a deliberate choice based on a combination of biological, practical, and economic considerations. Peptide synthesis cost and yield are highly dependent on length, with longer length becomes exponentially more expensive and suffers from lower yields and higher rates of synthesis errors [1]-[3]. Furthermore, the established antimicrobial peptide databases [4]-[7] reveals that this length range encompasses the majority of known natural and synthetic antimicrobial peptides, effectively capturing functional motifs while avoiding the synthetic challenges associated with very long sequences. Therefore, limiting sequences to certain length scope makes our approach financially feasible and practically executable, maximizing the chances of identifying active leads.

[1] The Current State of Peptide Drug Discovery: Back to the Future? *Journal of Medicinal Chemistry*, 61, 4.

[2] Computational Methods and Tools in Antimicrobial Peptide Research. *Journal of Chemical Information and Modeling*, 61, 3172-3196.

[3] The multifaceted nature of antimicrobial peptides: current synthetic chemistry approaches and future directions. *Chemical Society Reviews*, 50, 7820-7880.

[4] APD3: the antimicrobial peptide database as a tool for research and education. *Nucleic Acids Research*, 44, D1087-D1093.

[5] LAMP: A Database Linking Antimicrobial Peptides. *PLOS ONE* 8, e66557.

[6] CAMPR4: a database of natural and synthetic antimicrobial peptides. *Nucleic Acids Research*, 51, D377-D383.

[7] DBAASP v3: database of antimicrobial/cytotoxic activity and structure of peptides as a resource for development of new therapeutics. *Nucleic Acids Research*, 49, D1, 8.

11. Several comparisons lack details of statistical tests or corrections for multiple hypotheses, and raw activity data (IC₅₀/MIC tables, hemolysis datasets) have not been deposited in a public

repository.

Response: Thank you so much for this suggestion, and we have revised the manuscript to include details of statistical tests and corrections (Line 411-416, 430-432, 455-458). For raw activity data (MIC values, hemolysis data, and cytotoxicity data), they have been prepared as supplementary files (antimicrobial_activity_data.xlsx) of the manuscript and have also been submitted to the Github repository at <https://github.com/ChenSizhe13893461199/AMPLiT>. The hemolytic activity data have also been submitted to public database “Hemolytik2” and will be publicly available soon after the checking procedures by their website developing team. Thank you again for the valuable comment.

12. The manuscript should be thoroughly revised for grammar issues.

Response: Thank you so much for this kind remind, and we have thoroughly revised the grammar issues of the manuscript accordingly.

13. I suggest changing the title to “Ancient gut microbiomes harbor unexplored antimicrobial peptides”.

Response: We sincerely thank the reviewer for this insightful suggestion, and we have revised the manuscript title accordingly.

POINT-BY-POINT REPLY TO EDITORS AND REVIEWERS

Reviewer #1:

Remarks to the Author:

1. The existing biological mechanisms of encrypted peptides generation (post-translational cleavage; genomic mutations) cited in the authors' references are inconsistent with their computational approach. The custom script (ORF_hunter.py) defines ORFs solely through promoter and terminator predictions and applies a strict length cutoff (33-150 nt), which is inherently biased. Furthermore, the possibility that the AMPs reported in the cited literature occur naturally is reasonable. Still, the study in the manuscript appears to have mined short sequences that lack transcriptional and translational potential but exhibit moderate antibacterial activity, which may not reflect biological relevance. If these peptides do not correspond to naturally occurring or biologically active entities, their antimicrobial potency may not genuinely support the significance of this study.

Thank you so much for your critical comments and valuable suggestions, which are highly valuable for not only our current work but also for the advances in the whole field.

Since the proteomic data for ancient gut microbiomes are not available, the conventional approaches^{[1]-[2]} for identifying encrypted peptides do not apply to the ancient metagenome used in this study (Lines 174-183). Therefore, we chose to conduct the mentioned metagenome-based ORF identification as an initial filter to screen for potential candidates, and then we employed systematic biological assays to confirm the antimicrobial activity. We acknowledge that our method has some limitations, and it may not fully capture the complexity of certain encrypted peptides. We have discussed this limitation in the revised manuscript (Lines 94-95, lines 177-183), and we believe that further studies using multi-omics data may help fill this gap.

Besides, the mentioned selection of length cutoff (33-150 nt) was based on properties of most known AMPs in public dataset^{[3]-[4]} and was employed to narrow down the scope of AMP candidates with considerations of downstream chemical synthesis applicability and translational potentials. It may miss certain AMP candidates beyond the scope, and we have highlighted this in the revised manuscript (Lines 177-183, lines 209-212).

We agree with the reviewer that these mined short sequences may lack transcriptional and translational potential. To mitigate potential biases and enhance biological relevance, we implemented strict downstream filtering (Lines 94-97, manuscript Fig.1g), including prevalences across samples (occurred in at least 5 out of 6 ancient metagenomic samples), genomic context investigation, consistent conservation, and *in vivo/in vitro* experimental validations, co-supporting the potential biological significance of these AMPs. Additionally, we conducted Frequency of

optimal codons (FOP) and Codon Adaptation Index (CAI) Analysis [5]-[7] to evaluate the codon usage compatibility. Among these *S. copri*-derived AMP ORFs, only limited ORFs exhibit low FOP (<0.6) and CAI (<0.6) indexes, indicating that the codon usage among most of these ORFs resembles the patterns of the expressed host genes (annotated functional genes from *S. copri* genome NCBI Ref Seq: GCF_025151535.1) and implies their potential for transcription and translation. These results further supported the transcriptional and translational potential of our identified AMPs, but studies using multi-omics are needed to further confirm this. We have acknowledged this limitation in the revised manuscript (Lines 180-183).

Thank you so much for prompting a deeper consideration of our work's scope and implications. We explicitly acknowledged that more efforts (e.g. multi-omics approaches) are needed to further reveal the natural occurrence of these peptides in the future. Accordingly, these aspects mentioned by the reviewer have also been discussed in the manuscript to improve the clarity (Lines 174-191). We strongly believe that these opinions by the reviewer is very important not only for our study, but also for the advances in the relevant field.

References:

- [1] Molecular de-extinction of ancient antimicrobial peptides enabled by machine learning, *Cell Host & Microbe*, 31, 8, 1260-1274.e6
- [2] Deep-learning-enabled antibiotic discovery through molecular de-extinction. *Nature Biomedical Engineering* 8, 854-871
- [3] The multifaceted nature of antimicrobial peptides: current synthetic chemistry approaches and future directions. *Chemical Society Reviews*, 50, 7820-7880
- [4] Computational Methods and Tools in Antimicrobial Peptide Research. *Journal of Chemical Information and Modeling*, 61, 3172-3196
- [5] The codon adaptation index-a measure of directional synonymous codon usage bias, and its potential applications. *Nucleic Acids Research*, 15 (3): 1281-1295.
- [6] Revisiting the codon adaptation index from a whole-genome perspective: analyzing the relationship between gene expression and codon occurrence in yeast using a variety of models". *Nucleic Acids Research*, 31 (8): 2242-2251.
- [7] The Codon Statistics Database: A Database of Codon Usage Bias. *Molecular Biology and Evolution*, 39, 8, msac157.

2. The authors' own data, showing most contigs are longer than 1 kb, provides a direct opportunity to validate the fragmented origin of these ORFs. Specifically for the AMPs listed in Figure 2, did the authors attempt to annotate the larger protein sequences encoded on these same assembled contigs?

Thank you so much for the valuable comments, and we sincerely agree with your opinions that the size of these ancient contigs provides feasibility to trace the origins of these ORFs. For the AMPs listed in Figure 2a, we have confirmed the phylogenetic origins of these ancient contigs in *S. copri* (left panel below). And, we further annotated the larger protein sequences encoded by these ancient contigs, with genomic information consistent with the results shown in manuscript Figure 2a (right panel below). We have added this into the revised manuscript (Lines 112-114; Figure S2).

We sincerely believe that these valuable suggestions from reviewer are important for improving the clarity of our current manuscript, and we hope our efforts can alleviate the concerns from reviewer.

3. The study's central contribution needs to be clarified. The authors should decide whether the primary focus is on demonstrating the mining capabilities of the AMPLiT tool itself, or on revealing the AMP potential of ancient gut microbiomes and *S. copri*. This focus should be clearly reflected in targeted modifications to the abstract and main text better to highlight the study's core novelty and findings.

We sincerely agree with your valuable opinions that clarifying the central contribution of our study is important. Based on your kind suggestions, the abstract and main text body have been restructured to focus more on the biological significance of using ancient metagenomic samples for novel AMPs discovery (Lines 36-46, lines 174-195). We have moved the description of AMPLiT to the methods part (Lines 237-246). We believe that these targeted modifications suggested by the reviewer have significantly sharpened the manuscript's quality.

Reviewer #2:

Remarks to the Author:

1. While I thank the authors for their efforts, I feel that some points were not adequately addressed in the rebuttal. Perhaps this is due to my lack of clarity, so I will attempt to clarify my concerns and be more specific. The authors answer to my concern #2 about using efficiency as a justification seems to misdirect. They claim that their tool is better because it uses more features, but that is not about efficiency. The sentence on L100 still claims "AMPLiT identified putative AMPs within one hour, an efficiency infeasible for previous large pipelines on standard hardware." This is simply not true and the references given do not support it.

We deeply thank the reviewer for this critical correction and for holding us to a high standard of accuracy. We sincerely apologize for our inaccurate claim regarding computational efficiency. We have deleted it and revised the text (Lines 87-88) to state: "*AMPLiT deployable on standard hardware was used to identify putative AMP candidates.*" We have also deleted unsuitable references in the revised manuscript.

We sincerely acknowledge that tools like MACREL are computationally rapid in the AMP prediction. Our intended point, which we didn't articulate clearly in previous response letter, was to position AMPLiT's primary strength not as a fast predictor, but as a tool that offers a practical balance between predictive performance and computational burdens, especially in comparison to recently proposed AMP model demands extensive GPU resources and training time ^[1] (e.g. they claimed that 11 days of training costs are required for AI model training step). We have discussed this in the revised manuscript (Lines 80-83, lines 243-246).

Lastly, we feel grateful for the reviewer's re-clarification and kind suggestions. We are grateful for the reviewer's rigorous feedback, which has been essential in improving not only our current study.

References:

[1] Discovery of antimicrobial peptides with notable antibacterial potency by an LLM-based foundation model. *Science Advances*, 11, eads893.

2. Concerns regarding the codes of ORF detection used in our work. Overall, I still feel that the authors have not adequately addressed my concerns. With respect to whether their ORF detection is correct (my previous concern #5), I again feel that the authors are answering a different point than the one I was making. Indeed, encrypted peptides can sometimes be found in larger proteins (and be active). Often active peptides can be "trimmed" down to an active core. It could have been interesting to run 6-frame translation on the input, but this is not what their code is doing. To expand on the example given previously, the following sequence codes for AMP10.387_945:

```
>GMSC10.SMORF.002_433_233_034
```

```
ATGCTCCTCGCTGGCGATGAGTTCCGCCCCCTCCTCAAGAAGGAAGGCCTGCTGACCC
GCGACCCCAGAATGAAGGAGCGCAAGAAGTACGGCTTGAAAGCAGCCCGTCGCGCAC
CCCAGTTCTCCAAGAGATAA
```

Their code extracts two ORFs from this sequence:

```
ATGAGTTCCGCCCCCTCCTCAAGAAGGAAGGCCTGCTGA
```

and

```
ATGAAGGAGCGCAAGAAGTACGGCTTGAAAGCAGCCCGTCGCGCACCCCAGTTCTCC
AAGAGATAA
```

The second one appears to be the shorter fragment that was discussed earlier, that matches the AMP10.387_945 sequence from AMPSphere. As a side-note: this behaviour (outputting embedded ORFs) explains the lower molecular weights observed in AMPLiT predictions compared to other databases, as many peptides will be truncated due to the presence of out-of-frame start codons (cf. the statements on L104+).

However, consider the variation below with a single base change:

```
>GMSC10.SMORF.002_433_233_034_variant
```

```
ATGCTCCTCGCTGGCGATGAGTTCCGCCCCCTCCTCAAGAAGGAAGGCCTGCTGACCC
GCGACCCCAGAATGAAGGAGCGCAAGAAGTACGGCTTGAAAGCAGCCCGTCGCGCAC
CCCAGTTCTCCAAGAGATGA
```

This changes the penultimate base from A to G, which should be a silent mutation as TAA and TGA are both STOP in the standard table. This sequence codes for the exact same peptide, but now their code fails to extract the second one because the string ATG in the last four bases triggers the start of a new ORF even though it is out of frame. The authors may wish to simply acknowledge that their code inadvertently misses some encrypted peptides due to the way ORFs are extracted, and that this could be improved in future versions.

We sincerely appreciate the reviewer for this insightful and constructive feedback regarding our ORF detection methodology. We acknowledge that our approach, in rare edge cases, may miss some encrypted peptides. We have highlighted this limitation in the revised manuscript (Lines 174-180). Beyond this, according to the kind suggestions from the reviewer, we have prepared an updated version of the ORF detection tool (6-frame translation approach) for utilization in our or other users' future works. The updated script now performs a comprehensive translation of the input sequence, and we have uploaded the relevant computational codes to the GitHub repository (<https://github.com/ChenSizhe13893461199/AMPLiT>) (“ORF_hunter_updated.py”) accordingly.

And we tested the “ORF_hunter_updated.py” on the given example from reviewer below:

```
>GMSC10.SMORF.002_433_233_034_variant
```

```
ATGCTCCTCGCTGGCGATGAGTTCCGCCCCCTCCTCAAGAAGGAAGGCCTGCTGACCC
GCGACCCCAGAATGAAGGAGCGCAAGAAGTACGGCTTGAAAGCAGCCCGTCGCGCAC
```

CCCAGTTCTCCAAGAGATGA

And the updated “ORF_hunter_updated.py” now enables the detection of all available ORF sequences, even with the conditions of penultimate base changing from A to G (as shown below).

Header	Protein Sequence	DNA Sequence
>GMSC10.SMORF.002_433_233_034_variant	LLAGDEFRLPKKKEGLLRDPRMKERKKYGLKAARRAPQFSKR	ATGCTCCTCGCTGGCGATGAGTTCGCCCCCTCTCAAGAAGGAAGGCGCTGACCCCGCACCCAGATGAAGGAGCGCAA
>GMSC10.SMORF.002_433_233_034_variant	KERKKYGLKAARRAPQFSKR	ATGAAGGAGCGCAAGAAGTACGGCTTGAAGCAGCCCGTCGCGCACCCAGTTCTCCAAGAGATGA
>GMSC10.SMORF.002_433_233_034_variant	SSAPSSRRKAC	ATGAGTTCGCCCCCTCTCAAGAAGGAAGGCGCTGCTGA

We appreciate the reviewer for highlighting this technical issue, and we will strictly follow the guidance proposed by the reviewer in our future work. The reviewer's meticulous analysis not only significantly contributes to improve our current work, but also provides a clear direction to enhance precision for our future research proposal.

3. The 3rd point in their answer is very interesting, but a complete speculation from their data.

We sincerely appreciate the reviewer for this valuable comment. The reviewer's professional suggestions are beneficial to improve our current manuscript and provide insightful guidance to our analysis workflows. Based on the important comments from reviewer, we further conduct analysis to alleviate the potential concerns.

Regarding the previously mentioned AMP1 sequence KERKKYGLKAARRAPQFSKR that seems to match a longer peptide in AMPSphere, here we strictly followed the technical suggestions from reviewer in aspects of ORF detection and analyzed the available 3 genes (as shown below) encoding the aforementioned longer peptide in AMPSphere.

1. GMSC10.SMORF.000_305_373_387
2. GMSC10.SMORF.001_959_964_431
3. GMSC10.SMORF.002_433_233_034

(https://ampsphere.big-data-biology.org/amp?accession=AMP10.387_945)

Analysis revealed that the aforementioned 3 genes were not found to be contained within or aligned

to any regions within the same contig harboring AMP1 ORF, indicating different genetic origins between the 3 genes and AMP1 ORF (Full details of the genomic information for AMP1 on the contig are shown above).

Further analysis also showed that the AMP1 ORF are genetically different from the 3 aforementioned genes (containing shorter ORF corresponding to AMP 1 peptide sequences), with divergent bases highlighted by yellow color (shown below). Based on these observations, we believe that our identified AMP1 is not a fragment of AMP10.387_945 but constitutes an independent AMP with a distinct genetic origin.

We sincerely hope our efforts can alleviate the concerns from reviewer. And we highly appreciate the reviewer for these valuable and kind comments, which have significant benefits for improving the quality of our manuscript and our future studies.

POINT-BY-POINT REPLY TO EDITORS AND REVIEWERS

Reviewer #1:

Remarks to the Author:

Although I appreciate the authors' efforts, I believe that my comments 1-3 were not sufficiently addressed in the rebuttal. For instance, regarding comment 3, the absence of a comparison of MIC values with Vancomycin and the weak activity of the discovered AMPs do not adequately support the claim of the biological significance of using ancient metagenomic samples for novel AMP discovery.

We sincerely appreciate reviewer 1 for proposing these critical comments and valuable suggestions, which are highly constructive for not only our current work but also for the advances in the whole field. We sincerely appreciate this important point and agree that the *in vitro* dose-response assay comparisons with clinical antibiotics such as Vancomycin are essential in our future work. To alleviate the potential concerns, we have incorporated a specific discussion of the mentioned limitations to avoid overinterpretations in the current manuscript (Lines 242-247) and highlighted the efforts that we need to improve in our future research (Lines 232-255). In this section, we clarify that the current findings primarily demonstrate the feasibility and novelty of mining ancient metagenomes, and we explicitly outline the necessary future steps for efficacy benchmarking and optimization.

We recognize that the antimicrobial activity observed at this stage requires further optimization for clinical translation. Nonetheless, we believe this early-stage work bridges evolutionary biology and antimicrobial discovery by illustrating how ancient gut microbiomes can serve as a novel resource for identifying unique peptide scaffolds in the fight against modern pathogens. We sincerely appreciate the reviewer 1 for proposing these insightful suggestions, which have strengthened the clarity and scope of our manuscript

Reviewer #2:

The authors have addressed my previous concerns. I think there could be more analysis on the point related to the evolution of the genes partially matching other previously reported peptides, but this does not affect the text (it is only mentioned in the rebuttal document) and I recognise it is outside the scope of the current manuscript.

We sincerely appreciate reviewer 2 for the positive feedback and constructive suggestions on our previous revisions. Regarding the comment on further analysis of gene evolution in relation to partially matching known peptides, we agree that this represents an interesting and valuable avenue for our future investigation. According to your valuable suggestions, we have taken note of the

relevant aspects for our future investigations and work, where evolutionary and comparative genomic analyses could further contextualize the origin, conservation, and functional divergence of these ancient antimicrobial peptide sequences. We thank the reviewer for this thoughtful input, which is highly beneficial to our future studies in this area.